# Efficient Training of Visual Transformers with Small Datasets

**Yahui Liu** [*]
University of Trento
Fondazione Bruno Kessler
yahui.liu@unitn.it

**Enver Sangineto**
University of Trento
enver.sangineto@unitn.it

**Wei Bi**
Tencent AI Lab
victoriabi@tencent.com

**Nicu Sebe**
University of Trento
niculae.sebe@unitn.it

**Bruno Lepri**
Fondazione Bruno Kessler
lepri@fbk.eu

**Marco De Nadai**
Fondazione Bruno Kessler
work@marcodena.it

## Abstract

Visual Transformers (VTs) are emerging as an architectural paradigm alternative to Convolutional networks (CNNs). Differently from CNNs, VTs can capture global relations between image elements and they potentially have a larger representation capacity. However, the lack of the typical convolutional inductive bias makes these models more data hungry than common CNNs. In fact, some local properties of the visual domain which are embedded in the CNN architectural design, in VTs should be learned from samples. In this paper, we empirically analyse different VTs, comparing their robustness in a small training set regime, and we show that, despite having a comparable accuracy when trained on ImageNet, their performance on smaller datasets can be largely different. Moreover, we propose an auxiliary self-supervised task which can extract additional information from images with only a negligible computational overhead. This task encourages the VTs to learn spatial relations within an image and makes the VT training much more robust when training data is scarce. Our task is used jointly with the standard (supervised) training and it does not depend on specific architectural choices, thus it can be easily plugged in the existing VTs. Using an extensive evaluation with different VTs and datasets, we show that our method can improve (sometimes dramatically) the final accuracy of the VTs. Our code is available at: https://github.com/yhlleo/VTs-Drloc.

## 1 Introduction

Visual Transformers (VTs) are progressively emerging architectures in computer vision as an alternative to standard Convolutional Neural Networks (CNNs), and they have already been applied to many tasks, such as image classification [17, 53, 61, 35, 58, 60, 33, 59], object detection [4, 66, 14], segmentation [50], tracking [36], image generation [30, 28] and 3D data processing [65], to mention a few. These architectures are inspired by the well known Transformer [55], which is the de facto standard in Natural Language Processing (NLP) [15, 45], and one of their appealing properties is the possibility to develop a unified information-processing paradigm for both visual and textual domains. A pioneering work in this direction is ViT [17], in which an image is split using a grid of non-overlapping patches, and each patch is linearly projected in the input embedding space, so obtaining a "token". After that, all the tokens are processed by a series of multi-head attention and feed-forward layers, similarly to how (word) tokens are processed in NLP Transformers.

---

[*]Work done as intern at the Tencent AI Lab.

35th Conference on Neural Information Processing Systems (NeurIPS 2021).

A clear advantage of VTs is the possibility for the network to use the attention layers to model global relations between tokens, and this is the main difference with respect to CNNs, where the receptive field of the convolutional kernels locally limits the type of relations which can be learned. However, this increased representation capacity comes at a price, which is the lack of the typical CNN inductive biases, based on exploiting the locality, the translation invariance and the hierarchical structure of visual information [35, 58, 60]. As a result, VTs need a lot of data for training, usually more than what is necessary to standard CNNs [17]. For instance, ViT is trained with JFT-300M [17], a (proprietary) huge dataset of 303 million (weakly) labeled high-resolution images, and performs worse than ResNets [24] with similar capacity when trained on ImageNet-1K ($\sim$ 1.3 million samples [48]). This is likely due to the fact that ViT needs to learn some local proprieties of the visual data using more samples than a CNN, while the latter embeds these properties in its architectural design [47].

To alleviate this problem, a second generation of VTs has very recently been independently proposed by different groups [61, 35, 58, 60, 59, 33, 28]. A common idea behind these works is to mix convolutional layers with attention layers, in such a way providing a local inductive bias to the VT. These hybrid architectures enjoy the advantages of both paradigms: attention layers model long-range dependencies, while convolutional operations can emphasize the local properties of the image content. The empirical results shown in most of these works demonstrate that these second-generation VTs can be trained on ImageNet outperforming similar-size ResNets on this dataset [61, 35, 58, 60, 59, 33]. However, it is still not clear what is the behaviour of these networks when trained on medium-small datasets. In fact, from an application point of view, most of the computer vision tasks cannot rely on (supervised) datasets whose size is comparable with (or larger than) ImageNet.

In this paper, we compare to each other different second-generation VTs by either training them from scratch or fine-tuning them on medium-small datasets, and we empirically show that, despite their ImageNet results are basically on par with each other, their classification accuracy with smaller datasets largely varies. We also compare VTs with same capacity ResNets, and we show that, in most cases, VTs can match the ResNet accuracy when trained with small datasets. Moreover, we propose to use an auxiliary self-supervised *pretext* task and a corresponding loss function to regularize training in a small training set or few epochs regime. Specifically, the proposed task is based on (unsupervised) learning the spatial relations between the output token embeddings. Given an image, we *densely* sample random pairs from the final embedding grid, and, for each pair, we ask the network to guess the corresponding geometric distance. To solve this task, the network needs to encode both local and contextual information in each embedding. In fact, without local information, embeddings representing different input image patches cannot be distinguished the one from the others, while, without contextual information (aggregated using the attention layers), the task may be ambiguous.

Our task is inspired by ELECTRA [12], in which the (NLP) pretext task is densely defined for each output embedding (Section 2). Clark et al. [12] show that their task is more *sample-efficient* than commonly used NLP pretext tasks, and this gain is particularly strong with small-capacity models or relatively smaller training sets. Similarly, we exploit the fact that an image is represented by a VT using multiple token embeddings, and we use their relative distances to define a localization task over a subset of all the possible embedding pairs. This way, *for a single image forward pass*, we can compare many embedding pairs with each other, and average our localization loss over all of them. Thus, our task is drastically different from those multi-crop strategies proposed, for instance, in SwAV [6], which need to independently forward each input patch through the network. Moreover, differently from "ordering" based tasks [42], we can define pairwise distances on a large grid without modeling all the possible permutations (more details in Section 2).

Since our auxiliary task is self-supervised, our *dense relative localization* loss ($\mathcal{L}_{drloc}$) does not require additional annotation, and we use it jointly with the standard (supervised) cross-entropy as a regularization of the VT training. $\mathcal{L}_{drloc}$ is very easy-to-be-reproduced and, despite this simplicity, it can largely boost the accuracy of the VTs, especially when the VT is either trained from scratch on a small dataset, or fine-tuned on a dataset with a large domain-shift with respect to the pretraining ImageNet dataset. In our empirical analysis, based on different training scenarios, a variable amount of training data and different VT architectures, $\mathcal{L}_{drloc}$ has *always* improved the results of the tested baselines, sometimes boosting the final accuracy of tens of points (and up to 45 points).

In summary, our main contributions are:

1. We empirically compare to each other different VTs, showing that their behaviour largely differs when trained with small datasets or few training epochs.

2. We propose a relative localization auxiliary task for VT training regularization.

3. Using an extensive empirical analysis, we show that this task is beneficial to speed-up training and improve the generalization ability of different VTs, independently of their specific architectural design or application task.

## 2 Related work

In this section, we briefly review previous work related to both VTs and self-supervised learning.

**Visual Transformers.** Despite some previous work in which attention is used inside the convolutional layers of a CNN [57, 26], the first fully-transformer architectures for vision are iGPT [8] and ViT [17]. The former is trained using a "masked-pixel" self-supervised approach, similar in spirit to the common masked-word task used, for instance, in BERT [15] and in GPT [45] (see below). On the other hand, ViT is trained in a supervised way, using a special "class token" and a classification head attached to the final embedding of this token. Both methods are computationally expensive and, despite their very good results when trained on huge datasets, they underperform ResNet architectures when trained from scratch using only ImageNet-1K [17, 8]. VideoBERT [51] is conceptually similar to iGPT, but, rather than using pixels as tokens, each frame of a video is holistically represented by a feature vector, which is quantized using an off-the-shelf pretrained video classification model. DeiT [53] trains ViT using distillation information provided by a pretrained CNN.

The success of ViT has attracted a lot of interest in the computer vision community, and different variants of this architecture have been recently used in many tasks [53, 50, 30, 11]. However, as mentioned in Section 1, the lack of the typical CNN inductive biases in ViT, makes this model difficult to train without using (very) large datasets. For this reason, very recently, a second-generation of VTs has focused on hybrid architectures, in which convolutions are used jointly with long-range attention layers [61, 35, 58, 60, 59, 33, 28]. The common idea behind all these works is that the sequence of the individual token embeddings can be shaped/reshaped in a geometric grid, in which the position of each embedding vector corresponds to a fixed location in the input image. Given this geometric layout of the embeddings, convolutional layers can be applied to neighboring embeddings, so encouraging the network to focus on local properties of the image. The main difference among these works concerns *where* the convolutional operation is applied (e.g., only in the initial representations [61] or in all the layers [35, 58, 60, 59, 33], in the token to query/key/value projections [58] or in the forward-layers [60, 33, 28], etc.). In most of the experiments of this paper, we use three state-of-the-art second-generation VTs for which there is a public implementation: T2T [61], Swin [35] and CvT [58]). For each of them, we select the model whose number of parameters is comparable with a ResNet-50 [24] (more details in Section 3 and Section 5). We do not modify the native architectures because the goal of this work is to propose a pretext task and a loss function which can be easily plugged in existing VTs.

Similarly to the original Transformer [55], in ViT, an (absolute) *positional embedding* is added to the representation of the input tokens. In Transformer networks, positional embedding is used to provide information about the token order, since both the attention and the (individual token based) feed-forward layers are permutation invariant. In [35, 59], *relative* positional embedding [49] is used, where the position of each token is represented relatively to the others. Generally speaking, positional embedding is a representation of the token position which is *provided as input* to the network. Conversely, our relative localization loss exploits the relative positions (of the final VT embeddings) as a *pretext task* to extract additional information without manual supervision.

**Self-supervised learning.** Reviewing the vast self-supervised learning literature is out of the scope of this paper. However, we briefly mention that self-supervised learning was first successfully applied in NLP, as a means to get supervision from text by replacing costly manual annotations with *pretext* tasks [37, 38]. A typical NLP pretext task consists in masking a word in an input sentence and asking the network to guess which is the masked token [37, 38, 15, 45]. ELECTRA [12] is a *sample-efficient* language model in which the masked-token pretext task is replaced by a discriminative task defined over all the tokens of the input sentence. Our work is inspired by this method, since we propose a pretext task which can be efficiently computed by densely sampling the final VT embeddings. However, while the densely supervised ELECTRA task is obtained by randomly replacing (word) tokens and using a pre-trained BERT model to generate plausible replacements, we do not need a pre-trained model and we do not replace input tokens, being our task based on predicting the

inter-token geometric distances. In fact, in NLP tasks, tokens are discrete and limited (e.g., the set of words of a specific-language dictionary), while image patches are "continuous" and highly variable, hence a replacement-based task is hard to use in a vision scenario.

In computer vision, common pretext tasks with still images are based on extracting two different views from the same image (e.g., two different crops) and then considering these as a pair of *positive* images, likely sharing the same semantic content [9]. Most current self-supervised computer vision approaches can be categorised in contrastive learning [54, 25, 9, 23, 52, 56, 18], clustering methods [3, 67, 29, 5, 1, 20, 6, 7], asymmetric networks [22, 10] and feature-decorrelation methods [19, 63, 2, 27]. While the aforementioned approaches are all based on ResNets, very recently, both [11] and [7] have empirically tested some of these methods with a ViT architecture [17].

One important difference of our proposal with respect to previous work, is that we do not propose a fully-self-supervised method, but we rather use self-supervision jointly with standard supervision (i.e., image labels) in order to regularize VT training, hence our framework is a *multi-task learning* approach [13]. Moreover, our dense relative localization loss is not based on positive pairs, and we do *not* use multiple views of the same image in the current batch, thus our method can be used with standard (supervised) data-augmentation techniques. Specifically, our pretext task is based on predicting the relative positions of pairs of tokens extracted from the same image.

Previous work using localization for self-supervision is based on predicting the input image rotation [21] or the relative position of *adjacent patches* extracted from the same image [16, 42, 43, 39]. For instance, in [42], the network should predict the correct permutation of a grid of $3 \times 3$ patches (in NLP, a similar, permutation based pretext task, is *deshuffling* [46]). In contrast, we do not need to extract multiple patches from the same input image, since we can efficiently use the final token embeddings (thus, we need a *single* forward and backward pass per image). Moreover, differently from previous work based on localization pretext tasks, our loss is *densely* computed between many random pairs of (non necessarily adjacent) token embeddings. For instance, a trivial extension of the ordering task proposed in [42] using a grid of $7 \times 7$ patches would lead to 49! possible permutations, which becomes intractable if modeled as a classification task. Finally, in [14], the position of a random query patch is used for the self-supervised training of a transformer-based object detector [4]. However, the localization loss used in [14] is specific for the final task (object localization) and the specific DETR architecture [4], while our loss is generic and can be plugged in any VT.

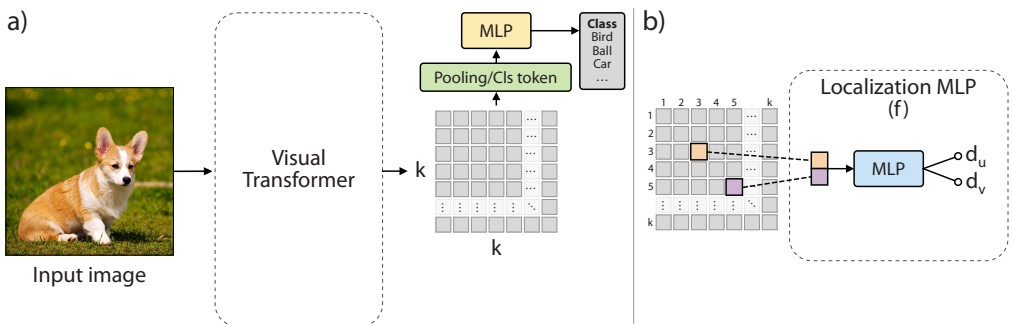

Figure 1: A schematic representation of the VT architecture. (a) A typical second-generation VT. (b) Our localization MLP which takes as input (concatenated) pairs of final token embeddings.

## 3  Preliminaries

A typical VT network takes as input an image split in a grid of (possibly overlapping) $K \times K$ patches. Each patch is projected in the input embedding space, obtaining a set of $K \times K$ input *tokens*. A VT is based on the Transformer multi-attention layers [55], which model pairwise relations over the token intermediate representations. Differently from a pure Transformer [55], the hybrid architectures mentioned in Section 1-2 usually shape or reshape the sequence of these token embeddings in a spatial grid, which makes it possible to apply convolutional operations over a small set of neighboring token embeddings. Using convolutions with a stride greater than 1 and/or pooling operations, the resolution of the initial $K \times K$ token grid can possibly be reduced, thus simulating the hierarchical structure of a CNN. We assume that the final embedding grid has a resolution of $k \times k$ (where, usually, $k \leq K$), see Fig. 1 (a).

The final $k \times k$ grid of embeddings represents the input image and it is used for the discriminative task. For instance, some methods include an additional "class token" which collects contextual information over the whole grid [17, 61, 58, 60, 59, 33], while others [35] apply an average global pooling over the final grid to get a compact representation of the whole image. Finally, a standard, small MLP head takes as input the whole image representation and it outputs a posterior distribution over the set of the target classes (Fig. 1 (a)). The VT is trained using a standard cross-entropy loss ($\mathcal{L}_{ce}$), computed using these posteriors and the image ground-truth labels.

When we plug our relative localization loss (Section 4) in an existing VT, we always use the native VT architecture of each tested method, without any change apart from the dedicated localization MLP (see Section 4). For instance, we use the class token when available, or the average pooling layer when it is not, and on top of these we use the cross-entropy loss. We also keep the positional embedding (Section 2) for those VTs which use it (see Section 4.1 for a discussion about this choice). The only architectural change we do is to downsample the final embedding grid of T2T [61] and CvT [58] to make them of the same size as that used in Swin [6]. Specifically, in Swin, the final grid has a resolution of $7 \times 7$ ($k = 7$), while, in T2T and in CvT, it is $14 \times 14$. Thus, in T2T and in CvT, we use a $2 \times 2$ average pooling (*without* learnable parameters) and we get a final $7 \times 7$ grid for all the three tested architectures. This pooling operation is motivated in Section 4.1, and it is used only with our localization task (it does not affect the posterior computed by the classification MLP). Finally, note that T2T uses convolutional operations only in the input stage, and it outputs a sequence of $14 \times 14 = 196$ embeddings, corresponding to its $14 \times 14$ input grid. In this case, we first reshape the sequence and then we use pooling. In the Supplementary Material, we show additional experiments with a ViT architecture [17], in which we adopt the same reshaping and pooling strategy.

## 4 Dense relative localization task

The goal of our regularization task is to encourage the VT to learn spatial information without using additional manual annotations. We achieve this by *densely* sampling *multiple* embedding pairs *for each image* and asking the network to guess their relative distances. In more detail, given an image $x$, we denote its corresponding $k \times k$ grid of final embeddings (Section 3), as $G_x = \{\mathbf{e}_{i,j}\}_{1 \leq i,j \leq k}$, where $\mathbf{e}_{i,j} \in \mathbb{R}^d$, and $d$ is the dimension of the embedding space. For each $G_x$, we randomly sample multiple pairs of embeddings and, for each pair $(\mathbf{e}_{i,j}, \mathbf{e}_{p,h})$, we compute the 2D normalized target translation offset $(t_u, t_v)^T$, where:

$$t_u = \frac{|i - p|}{k}, \quad t_v = \frac{|j - h|}{k}, \quad (t_u, t_v)^T \in [0, 1]^2. \tag{1}$$

The selected embedding vectors $\mathbf{e}_{i,j}$ and $\mathbf{e}_{p,h}$ are concatenated and input to a small MLP ($f$), with two hidden layers and two output neurons, one per spatial dimension (Fig. 1 (b)), which predicts the relative distance between position $(i, j)$ and position $(p, h)$ on the grid. Let $(d_u, d_v)^T = f(\mathbf{e}_{i,j}, \mathbf{e}_{p,h})^T$. Given a mini-batch $B$ of $n$ images, our *dense relative localization loss* is:

$$\mathcal{L}_{drloc} = \sum_{x \in B} \mathbb{E}_{(\mathbf{e}_{i,j}, \mathbf{e}_{p,h}) \sim G_x}[|(t_u, t_v)^T - (d_u, d_v)^T|_1]. \tag{2}$$

In Eq. 2, for each image $x$, the expectation is computed by sampling uniformly at random $m$ pairs $(\mathbf{e}_{i,j}, \mathbf{e}_{p,h})$ in $G_x$, and averaging the $L_1$ loss between the corresponding $(t_u, t_v)^T$ and $(d_u, d_v)^T$.

$\mathcal{L}_{drloc}$ is added to the standard cross-entropy loss ($\mathcal{L}_{ce}$) of each native VT (Section 3). The final loss is: $\mathcal{L}_{tot} = \mathcal{L}_{ce} + \lambda \mathcal{L}_{drloc}$. We use $\lambda = 0.1$ in all the experiments with both T2T and CvT, and $\lambda = 0.5$ in case of Swin. Note that the same pairwise localization task can be associated with slightly different loss formulations. In the Supplementary Material we present some of these variants and we compare them empirically with each other.

### 4.1 Discussion

Intuitively, $\mathcal{L}_{drloc}$ transforms the relative positional embedding (Section 2), used, for instance, in Swin [35], in a pretext task, asking the network to guess which is the relative distance of a random subset of all the possible token pairs. Thus a question arises: is the relative positional embedding used in some VTs sufficient for the localization MLP ($f$) to solve the localization task? The experiments presented in Section 5.2-5.3 show that, when we plug $\mathcal{L}_{drloc}$ on CvT, in which *no kind* of positional

embedding is used [58], the relative accuracy boost is usually *smaller* than in case of Swin, confirming that the relative positional embedding, used in the latter, is not sufficient to make our task trivial. We further analyze this point in the Supplementary Material.

In Section 3, we mentioned that, in case of T2T and CvT, we average-pool the final grid and we obtain a $7 \times 7$ grid $G_x$. In fact, in preliminary experiments with both T2T and CvT at their original $14 \times 14$ resolution, we observed a very slow convergence of $\mathcal{L}_{drloc}$. We presume this is due to the fact that, with a finer grid, the localization task is harder. This slows down the convergence of $f$, and it likely generates noisy gradients which are backpropagated through the whole VT (see also the Supplementary Material). We leave this for future investigation and, in the rest of this article, we always assume that our pretext task is computed with a $7 \times 7$ grid $G_x$.

Table 1: The size of the datasets used in our empirical analysis.

| Dataset | Train size | Test size | Classes |
|---|---|---|---|
| ImageNet-1K [48] | 1,281,167 | 100,000 | 1000 |
| ImageNet-100 [52] | 126,689 | 5,000 | 100 |
| CIFAR-10 [31] | 50,000 | 10,000 | 10 |
| CIFAR-100 [31] | 50,000 | 10,000 | 100 |
| Oxford Flowers102 [41] | 2,040 | 6,149 | 102 |
| SVHN [40] | 73,257 | 26,032 | 10 |
| *DomainNet* ClipArt | 33,525 | 14,604 | |
| Infograph | 36,023 | 15,582 | |
| Painting | 50,416 | 21,850 | 345 |
| Quickdraw | 120,750 | 51,750 | |
| Real | 120,906 | 52,041 | |
| Sketch | 48,212 | 20,916 | |

## 5 Experiments

All the experiments presented in this section are based on image classification tasks, while in the Supplementary Material we also show object detection, instance segmentation and semantic segmentation tasks. In this section we use 11 different datasets: ImageNet-100 (IN-100) [52, 56], which is a subset of 100 classes of ImageNet-1K [48]; CIFAR-10 and CIFAR-100 [31], Oxford Flowers102 [41] and SVHN [40], which are four widely used computer vision datasets; and the six datasets of DomainNet [44], a benchmark commonly used for domain adaptation tasks. We chose the latter because of the large domain-shift between some of its datasets and ImageNet-1K, which makes the fine-tuning experiments non-trivial. Tab. 1 shows the size of each dataset.

We used, when available, the official VT code (for T2T [61] and Swin [35]) and a publicly available implementation of CvT [58][2]. In the fine-tuning experiments (Section 5.3), we use only T2T and Swin because of the lack of publicly available ImageNet pre-trained CvT networks. For each of the three baselines, we chose a model of comparable size to ResNet-50 (25M parameters): see Tab. 3 for more details. In the Supplementary Material, we show additional results obtained with larger models (ViT-B [17]), larger datasets (e.g., ImageNet-1K) and more training epochs. When we plug our loss on one of the adopted baselines, we follow Section 4, *keeping unchanged* the VT architecture apart from our localization MLP ($f$). Moreover, in all the experiments, we train the baselines, both with and without our localization loss, using the same data-augmentation protocol for all the models, and we use the VT-specific hyper-parameter configuration suggested by the authors of each VT. We do *not* tune the VT-specific hyperparameters when we use our loss and we keep fixed the values of $m$ and $\lambda$ (Section 5.1) in all the experiments. We train each model using 8 V100 32GB GPUs.

### 5.1 Ablation study

In Tab. 2 (a) we analyze the impact on the accuracy of different values of $m$ (the total number of embedding pairs used per image, see Section 4). Since we use the same grid resolution for all the VTs (i.e., $7 \times 7$, Section 3), also the maximum number of possible embeddings per image is the same for all the VTs ($k^2 = 49$). Using the results of Tab. 2 (a) (based on CIFAR-100 and Swin), we chose

---

[2]https://github.com/lucidrains/vit-pytorch

$m = 64$ for all the VTs and all the datasets. Moreover, Tab. 2 (b) shows the influence of the loss weight $\lambda$ (Section 4) for each of the three baselines, which motivates our choice of using $\lambda = 0.1$ for both CvT and T2T and $\lambda = 0.5$ for Swin.

These values of $m$ and $\lambda$ *are kept fixed* in all the other experiments of this paper, independently of the dataset, the main task (e.g., classification, detection, segmentation, etc.), and the training protocol (from scratch or fine-tuning). This is done to emphasise the ease of use of our loss. Finally, in the Supplementary Material, we analyze the influence of the size of the localization MLP ($f$).

Table 2: CIFAR-100, 100 training epochs: (a) the influence on the accuracy of the number of pair samples ($m$) in $L_{drloc}$ using Swin, and (b) the influence of the $\lambda$ value using all the 3 VT baselines.

<div>
(a)

| Model | Top-1 Acc. |
|---|---|
| A: Swin-T | 53.28 |
| B: A + $\mathcal{L}_{drloc}$, $m$=32 | 63.70 |
| C: A + $\mathcal{L}_{drloc}$, $m$=64 | **66.23** |
| D: A + $\mathcal{L}_{drloc}$, $m$=128 | 65.16 |
| E: A + $\mathcal{L}_{drloc}$, $m$=256 | 64.87 |

(b)

| Model | $\lambda$=0.0 | $\lambda$=0.1 | $\lambda$=0.5 | $\lambda$=1.0 |
|---|---|---|---|---|
| CvT-13 | 73.50 | **74.51** | 74.07 | 72.84 |
| Swin-T | 53.28 | 58.15 | **66.23** | 64.28 |
| T2T-ViT-14 | 65.16 | **68.03** | 67.03 | 66.53 |
</div>

Table 3: Top-1 accuracy on IN-100 using either 100 or 300 epochs. In the former case, we show the average and the standard deviation values obtained by repeating each single experiment 5 times with 5 different random seeds.

| | Model | # Params | ImageNet-100 | |
|---|---|---|---|---|
| | | (M) | 100 epochs | 300 epochs |
| CvT | CvT-13 | 20 | $85.62 \pm 0.05$ | 90.16 |
| | CvT-13+$\mathcal{L}_{drloc}$ | 20 | **86.09** $\pm 0.12$ (+0.47) | **90.28** (+0.12) |
| Swin | Swin-T | 29 | $82.66 \pm 0.10$ | 89.68 |
| | Swin-T+$\mathcal{L}_{drloc}$ | 29 | **83.95** $\pm 0.05$ (+1.29) | **90.32** (+0.64) |
| T2T | T2T-ViT-14 | 22 | $82.67 \pm 0.01$ | 87.76 |
| | T2T-ViT-14+$\mathcal{L}_{drloc}$ | 22 | **83.74** $\pm 0.08$ (+1.07) | **88.16** (+0.40) |

## 5.2 Training from scratch

In this section, we analyze the performance of both the VT baselines and our regularization loss using small-medium datasets and different number of training epochs, simulating a scenario with limited computational resources and/or limited training data. In fact, while fine-tuning a model pre-trained on ImageNet-1K is the most common protocol when dealing with small training datasets, this is not possible when, e.g., the network input is not an RGB image (e.g., in case of 3D point cloud data [65]) or when using a task-specific backbone architecture [32, 34]. In these cases, the network needs to be trained from scratch on the target dataset, thus, investigating the robustness of the VTs when trained from scratch with relatively small datasets, is useful for those application domains in which a fine-tuning protocol cannot be adopted.

We start by analyzing the impact on the accuracy of the number of training epochs on IN-100. Tab. 3 shows that, using $\mathcal{L}_{drloc}$, *all the tested VTs* show an accuracy improvement, and this boost is larger with fewer epochs. As expected, our loss acts as a regularizer, whose effects are more pronounced in a shorter training regime. We believe this result is particularly significant considering the larger computational times which are necessary to train typical VTs with respect to ResNets.

In Tab. 4, we use all the other datasets and we train from scratch with 100 epochs (see the Supplementary Material for longer training protocols). First, we note that the accuracy of the VT baselines varies a lot depending on the dataset (which is expected), but also depending on the specific VT architecture. This is largely in contrast with the ImageNet-1K results, where the difference between the three baselines is much smaller. As a reference, when these VTs are trained on ImageNet-1K (for 300 epochs), the differences of their respective top-1 accuracy is much smaller: Swin-T, 81.3 [35]; T2T-ViT-14, 81.5 [61]; CvT-13, 81.6 [58]. Conversely, Tab. 4 shows that, for instance, the accuracy difference between CvT and Swin is about 45-46 points in Quickdraw and Sketch, 30 points

on CIFAR-10, and about 20 points on many other datasets. Analogously, the difference between CvT and T2T is between 20 and 25 points in Sketch, Painting and Flowers102, and quite significant in the other datasets. This comparison shows that CvT is usually much more robust in a small training set regime with respect to the other two VTs, a behaviour which is completely hidden when the training/evaluation protocol is based on large datasets only.

In the same table, we also show the accuracy of these three VTs when training is done using $\mathcal{L}_{drloc}$ as a regularizer. Similarly to the IN-100 results, also in this case our loss *improves the accuracy of all the tested VTs in all the datasets*. Most of the time, this improvement is quite significant (e.g., almost 4 points on SVHN with CvT), and sometimes dramatic (e.g., more than 45 points on Quickdraw with Swin). These results show that a self-supervised auxiliary task can provide a significant "signal" to the VT when the training set is limited, and, specifically, that our loss can be very effective in boosting the accuracy of a VT trained from scratch in this scenario.

In Tab. 4 we also report the results we obtained using a ResNet-50, trained with 100 epochs and the standard ResNet training protocol (e.g., using Mixup [64] and CutMix [62] data-augmentations, etc.). These results show that the best performing VT (CvT) is usually comparable with a same size ResNet, and demonstrate that VTs can potentially be trained from scratch with darasets smaller than InageNet-1K. Finally, in the last row of the same table, we train the ResNet-50 baseline jointly with our pretext task. In more detail, we replace the VT token embedding grid ($G_x$ in Eq. 2) with the last convolutional feature map of the ResNet, and we apply our loss (Eq. 2) on top of this map. A comparison between the results of the last 2 rows of Tab. 4 shows that our loss is useful also when used with a ResNet (see the Supplementary Material for longer training protocols). When using ResNets, the improvement obtained with our loss is marginal, but it is consistent in 9 out of 10 datasets. The smaller improvement with respect to the analogous VT results may probably be explained by the fact that ResNets already embed local inductive biases in their architecture, thus a localization auxiliary task is less helpful (Section 1).

Table 4: Top-1 accuracy of VTs and ResNets, trained from scratch on different datasets (100 epochs).

| | | CIFAR-10 | CIFAR-100 | Flowers102 | SVHN | ClipArt | Infograph | Painting | Quickdraw | Real | Sketch |
|---|---|---|---|---|---|---|---|---|---|---|---|
| CvT | CvT-13 | 89.02 | 73.50 | 54.29 | 91.47 | 60.34 | 19.39 | 54.79 | 70.10 | 76.33 | 56.98 |
| | CvT-13+$\mathcal{L}_{drloc}$ | **90.30** | **74.51** | **56.29** | **95.36** | **60.64** | **20.05** | **55.26** | **70.36** | **77.05** | **57.56** |
| | | (+1.28) | (+1.01) | (+2.00) | (+3.89) | (+0.30) | (+0.67) | (+0.47) | (+0.26) | (+0.68) | (+0.58) |
| Swin | Swin-T | 59.47 | 53.28 | 34.51 | 71.60 | 38.05 | 8.20 | 35.92 | 24.08 | 73.47 | 11.97 |
| | Swin-T+$\mathcal{L}_{drloc}$ | **83.89** | **66.23** | **39.37** | **94.23** | **47.47** | **10.16** | **41.86** | **69.41** | **75.59** | **38.55** |
| | | (+24.42) | (+12.95) | (+4.86) | (+22.63) | (+9.42) | (+1.96) | (+5.94) | (+45.33) | (+2.12) | (+26.58) |
| T2T | T2T-ViT-14 | 84.19 | 65.16 | 31.73 | 95.36 | 43.55 | 6.89 | 34.24 | 69.83 | 73.93 | 31.51 |
| | T2T-ViT-14+$\mathcal{L}_{drloc}$ | **87.56** | **68.03** | **34.35** | **96.49** | **52.36** | **9.51** | **42.78** | **70.16** | **74.63** | **51.95** |
| | | (+3.37) | (+2.87) | (+2.62) | (+1.13) | (+8.81) | (+2.62) | (+8.54) | (+0.33) | (+0.70) | (+20.44) |
| ResNet | ResNet-50 | 91.78 | 72.80 | 46.92 | 96.45 | 63.73 | 19.81 | 53.22 | 71.38 | 75.28 | **60.08** |
| | ResNet-50+$\mathcal{L}_{drloc}$ | **92.03** | **72.94** | **47.65** | **96.53** | **63.93** | **20.79** | **53.52** | **71.57** | **75.56** | 59.62 |
| | | (+0.25) | (+0.14) | (+0.73) | (+0.08) | (+0.20) | (+0.98) | (+0.30) | (+0.19) | (+0.28) | (-0.46) |

## 5.3 Fine-tuning

In this section, we analyze a typical fine-tuning scenario, in which a model is pre-trained on a big dataset (e.g., ImageNet), and then fine-tuned on the target domain. Specifically, in *all* the experiments, we use VT models pre-trained by the corresponding VT authors on ImageNet-1K *without* our localization loss. The difference between the baselines and ours concerns *only* the fine-tuning stage, which is done in the standard way for the former and using our $\mathcal{L}_{drloc}$ regularizer for the latter. Starting from standard pre-trained models and using our loss only in the fine-tuning stage, emphasises the easy to use of our proposal in practical scenarios, in which fine-tuning can be done without re-training the model on ImageNet. As mentioned in Section 5, in this analysis we do not include CvT because of the lack of publicly available ImageNet-1K pre-trained models for this architecture.

Table 5: Pre-training on ImageNet-1K and then fine-tuning on the target dataset (top-1 accuracy, 100 fine-tuning epochs).

| | | CIFAR-10 | CIFAR-100 | Flowers102 | SVHN | ClipArt | Infograph | Painting | Quickdraw | Real | Sketch |
|---|---|---|---|---|---|---|---|---|---|---|---|
| Swin | Swin-T | 97.95 | 88.22 | 98.03 | 96.10 | 73.51 | 41.07 | 72.99 | 75.81 | 85.48 | 72.37 |
| | Swin-T+$\mathcal{L}_{drloc}$ | **98.37** | **88.40** | **98.21** | **97.87** | **79.51** | **46.10** | **73.28** | **76.01** | **85.61** | **72.86** |
| | | (+0.42) | (+0.18) | (+0.18) | (+1.77) | (+6.00) | (+5.03) | (+0.29) | (+0.20) | (+0.13) | (+0.49) |
| T2T | T2T-ViT-14 | 98.37 | 87.33 | 97.98 | 97.03 | 74.59 | 38.53 | 72.29 | 74.16 | 84.56 | 72.18 |
| | T2T-ViT-14+$\mathcal{L}_{drloc}$ | **98.52** | **87.65** | **98.08** | **98.20** | **78.22** | **45.69** | **72.42** | **74.27** | **84.57** | **72.29** |
| | | (+0.15) | (+0.32) | (+0.10) | (+1.17) | (+3.63) | (+7.16) | (+0.13) | (+0.11) | (+0.01) | (+0.11) |
| ResNet | ResNet-50 | 97.65 | 85.44 | 96.59 | 96.60 | 75.22 | 44.30 | 66.58 | 72.12 | 80.40 | 67.77 |
| | ResNet-50+$\mathcal{L}_{drloc}$ | **97.74** | **85.65** | **96.72** | **96.71** | **75.51** | **44.39** | **69.03** | **72.21** | **80.54** | **68.14** |
| | | (+0.09) | (+0.21) | (+0.13) | (+0.11) | (+0.29) | (+0.09) | (+2.45) | (+0.09) | (+0.14) | (+0.37) |

The results are presented in Tab. 5. Differently from the results shown in Section 5.2, the accuracy difference between the T2T and Swin baselines is much less pronounced, and the latter outperforms the former in most of the datasets. Moreover, analogously to all the other experiments, using $\mathcal{L}_{drloc}$ leads to an accuracy improvement *with all the tested VTs and in all the datasets*. For instance, on Infograph, Swin with $\mathcal{L}_{drloc}$ improves of more than 5 points, and T2T more than 7 points.

In the last two rows of Tab. 5, we show the ResNet based results. The comparison between ResNet and the VT baselines shows that the latter are very competitive in this fine-tuning scenario, even more than with a training-from-scratch protocol (Tab. 4). For instance, the two VT baselines (without our loss) are outperformed by ResNet only in 2 out of 10 datasets. This confirms that VTs are likely to be widely adopted in computer vision applications in the near future, independently of the training set size. Finally, analogously to the experiments in Section 5.2, Tab. 5 shows that our loss is (marginally) helpful also in ResNet fine-tuning.

## 6 Conclusion

In this paper, we have empirically analyzed different VTs, showing that their performance largely varies when trained with small-medium datasets, and that CvT is usually much more effective in generalizing with less data. Moreover, we proposed a self-supervised auxiliary task to regularize VT training. Our localization task, inspired by [12], is densely defined for a random subset of final token embedding pairs, and it encourages the VT to learn spatial information.

In our extensive empirical analysis, with 11 datasets, different training scenarios and three VTs, our dense localization loss *has always improved the corresponding baseline accuracy*, usually by a significant margin, and sometimes dramatically (up to +45 points). We believe that this shows that our proposal is an easy-to-reproduce, yet very effective tool to boost the performance of VTs, especially in training regimes with a limited amount of data/training time. It also paves the way to investigating other forms of self-supervised/multi-task learning which are specific for VTs, and can help VT training without resorting to the use of huge annotated datasets.

**Limitations.** A deeper analysis on why fine-grained embedding grids harm our auxiliary task (Section 4.1) was left as future work. Moreover, while we show in the Supplementary Material a few experiments with ViT-B, which confirm the usefulness of $\mathcal{L}_{drloc}$ when used with bigger VT models, in our analysis we mainly focused on VTs of approximately the same size as a ResNet-50. The goal of this paper is to investigate the VT behaviour with medium-small datasets, thus, high-capacity models most likely are not the best choice in a training scenario with scarcity of data.

## Acknowledgements

This work was partially supported by the EU H2020 AI4Media No. 951911 project and by the EUREGIO project OLIVER.

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
