# Efficient Training of Visual Transformers with Small Datasets - Supplementary Material

**Yahui Liu** *
University of Trento
Fondazione Bruno Kessler
yahui.liu@unitn.it

**Enver Sangineto**
University of Trento
enver.sangineto@unitn.it

**Wei Bi**
Tencent AI Lab
victoriabi@tencent.com

**Nicu Sebe**
University of Trento
niculae.sebe@unitn.it

**Bruno Lepri**
Fondazione Bruno Kessler
lepri@fbk.eu

**Marco De Nadai**
Fondazione Bruno Kessler
work@marcodena.it

## 1 Pseudocode of the dense relative localization task

In order to emphasise the simplicity and the ease of reproduction of our proposed method, in Figure 2 we show a PyTorch-like pseudocode of our auxiliary task with the associated $\mathcal{L}_{drloc}$ loss.

```python
# n        : batch size
# m        : number of pairs
# k X k    : resolution of the embedding grid
# D        : dimension of each token embedding
# x        : a tensor of n embedding grids, shape=[n, D, k, k]

def position_sampling(k, m, n):
    pos_1 = torch.randint(k, size=(n, m, 2))
    pos_2 = torch.randint(k, size=(n, m, 2))
    return pos_1, pos_2

def collect_samples(x, pos, n):
    _, c, h, w = x.size()
    x = x.view(n, c, -1).permute(1, 0, 2).reshape(c, -1)
    pos = ((torch.arange(n).long().to(pos.device) * h * w).view(n, 1)
        + pos[:, :, 0] * h + pos[:, :, 1]).view(-1)
    return (x[:, pos]).view(c, n, -1).permute(1, 0, 2)

def dense_relative_localization_loss(x):
    n, D, k, k = x.size()
    pos_1, pos_2 = position_sampling(k, m, n)

    deltaxy = abs((pos_1 - pos_2).float()) # [n, m, 2]
    deltaxy /= k

    pts_1 = collect_samples(x, pos_1, n).transpose(1, 2) # [n, m, D]
    pts_2 = collect_samples(x, pos_2, n).transpose(1, 2) # [n, m, D]
    predxy = MLP(torch.cat([pts_1, pts_2], dim=2))
    return L1Loss(predxy, deltaxy)
```

Figure 2: A PyTorch-like pseudocode of our dense relative localization task and the corresponding $\mathcal{L}_{drloc}$ loss.

---

*Work done as intern at the Tencent AI Lab.

35th Conference on Neural Information Processing Systems (NeurIPS 2021).

## 2 Loss Variants

In this section, we present different loss function variants associated with our relative localization task, which are empirically evaluated in Sec. 2.1. The goal is to show that the auxiliary task proposed in the main paper can be implemented in different ways and to analyze the differences between these implementations.

The first variant consists in including negative target offsets:

$$t'_u = \frac{i - p}{k}, \quad t'_v = \frac{j - h}{k}, \quad (t'_u, t'_v)^T \in [-1, 1]^2. \tag{3}$$

Replacing $(t_u, t_v)^T$ in Eq. 2 in the main paper with $(t'_u, t'_v)^T$ computed as in Eq. 3, and keeping all the rest unchanged, we obtain the first variant, which we call $\mathcal{L}^*_{drloc}$.

In the second variant, we transform the regression task in Eq. 2 in the main paper in a classification task, and we replace the $L_1$ loss with the cross-entropy loss. In more detail, we use as target offsets:

$$c_u = i - p, \quad c_v = j - h, \quad (c_u, c_v)^T \in \{-k, ..., k\}^2, \tag{4}$$

and we associate each of the $2k+1$ discrete elements in $C = \{-k, ..., k\}$ with a "class". Accordingly, the localization MLP $f$ is modified by replacing the 2 output neurons with 2 different sets of neurons, one per spatial dimension ($u$ and $v$). Each set of neurons represents a discrete offset prediction over the $2k + 1$ "classes" in $C$. Softmax is applied *separately* to each set of $2k + 1$ neurons, and the output of $f$ is composed of two posterior distributions over $C$: $(\mathbf{p}_u, \mathbf{p}_v)^T = f(\mathbf{e}_{i,j}, \mathbf{e}_{p,h})^T$, where $\mathbf{p}_u, \mathbf{p}_v \in [0, 1]^{2k+1}$. Eq. 2 in the main paper is then replaced by:

$$\mathcal{L}^{ce}_{drloc} = -\sum_{x \in B} \mathbb{E}_{(\mathbf{e}_{i,j}, \mathbf{e}_{p,h}) \sim G_x}[log(\mathbf{p}_u[c_u]) + log(\mathbf{p}_v[c_v])], \tag{5}$$

where $\mathbf{p}_u[c_u]$ indicates the $c_u$-th element of $\mathbf{p}_u$ (and similarly for $\mathbf{p}_v[c_v]$).

Note that, using the cross-entropy loss in Eq. 5, corresponds to considering $C$ an unordered set of "categories". This implies that prediction errors in $\mathbf{p}_u$ (and $\mathbf{p}_v$) are independent of the "distance" with respect to the ground-truth $c_u$ (respectively, $c_v$). In order to alleviate this problem, and inspired by [5], in the third variant we propose, we impose a Gaussian prior on $\mathbf{p}_u$ and $\mathbf{p}_v$, and we minimize the normalized squared distance between the expectation of $\mathbf{p}_u$ and the ground-truth $c_u$ (respectively, $\mathbf{p}_v$ and $c_v$). In more detail, let $\mu_u = \sum_{c \in C} \mathbf{p}_u[c] * c$ and $\sigma^2_u = \sum_{c \in C} \mathbf{p}_u[c] * (c - \mu_u)^2$ (and similarly for $\mu_v$ and $\sigma^2_v$). Then, Eq. 5 is replaced by:

$$\mathcal{L}^{reg}_{drloc} = \sum_{x \in B} \mathbb{E}_{(\mathbf{e}_{i,j}, \mathbf{e}_{p,h}) \sim G_x}\left[\frac{(c_u - \mu_u)^2}{\sigma^2_u} + \alpha log(\sigma_u) + \frac{(c_v - \mu_v)^2}{\sigma^2_v} + \alpha log(\sigma_v)\right], \tag{6}$$

where the terms $log(\sigma_u)$ and $log(\sigma_v)$ are used for variance regularization and $\alpha$ weights the importance of the Gaussian prior [5]. In preliminary experiments in which we tuned the $\alpha$ parameter using Swin, we found that the default value of $\alpha = 0.001$, as suggested in [5], works well in our scenario, thus we adopted it for all the experiments involving $\mathcal{L}^{reg}_{drloc}$.

The fourth variant we propose is based on a "very-dense" localization loss, where $\mathcal{L}_{drloc}$ is computed *for every transformer block* of VT. Specifically, let $G^l_x$ be the $k_l \times k_l$ grid of token embeddings produced by the $l$-th block of the VT, and let $L$ be the total number of these blocks. Then, Eq. 2 in the main paper is replaced by:

$$\mathcal{L}^{all}_{drloc} = \sum_{x \in B} \sum_{l=1}^{L} \mathbb{E}_{(\mathbf{e}_{i,j}, \mathbf{e}_{p,h}) \sim G^l_x}[|(t^l_u, t^l_v)^T - (d^l_u, d^l_v)^T|_1], \tag{7}$$

where $(t^l_u, t^l_v)^T$ and $(d^l_u, d^l_v)^T$ are, respectively, the target (see main paper Eq. 1) and the prediction offsets computed at block $l$ using the randomly sampled pair $(\mathbf{e}_{i,j}, \mathbf{e}_{p,h}) \in G^l_x$. For each block, we use a block-specific MLP $f^l$ to compute $(d^l_u, d^l_v)^T$. Note that, using Eq. 7, the initial layers of VT receive more "signal", because each block $l$ accumulates the gradients produced by all the blocks $l' \geq l$.

Apart from $\mathcal{L}^{all}_{drloc}$, all the other proposed variants are very computationally efficient, because they involve only one forward and one backward pass per image, and $m$ forward passes through $f$.

## 2.1 Empirical comparison of the loss variants

In Tab. 6, we compare the loss variants with each other, where the baseline model is Swin [8] (row (A)). For these experiments, we use IN-100, we train all the models for 100 epochs, and, as usual, we show the top-1 classification accuracy on the test set.

When we plug $\mathcal{L}_{drloc}$ on top of Swin (main paper, Sec. 4), the final accuracy increases by 1.26 points (B). All the other dense localization loss variants underperform $\mathcal{L}_{drloc}$ (C-F). A bit surprisingly, the very-dense localization loss $\mathcal{L}_{drloc}^{all}$ is significantly outperformed by the much simpler (and computationally more efficient) $\mathcal{L}_{drloc}$. Moreover, $\mathcal{L}_{drloc}^{all}$ is the only variant which underperforms the baseline. We presume that this is due to the fact that most of the Swin intermediate blocks have resolution grids $G_x^l$ finer than the last grid $G_x^L$ ($l < L$, $k_l > k_L$, Sec. 2), and this makes the localization task harder, slowing down the convergence of $f^l$, and likely providing noisy gradients to the VT (see the discussion in the main paper, Sec. 4.1). In all the other experiments (both in the main paper and in this Supplementary Material), we always use $\mathcal{L}_{drloc}$ as the relative localization loss.

Table 6: IN-100, 100 epoch training: a comparison between different loss variants.

|    | Model | Top-1 Acc. |
|----|-------|------------|
| A: | Swin-T | 82.76 |
| B: | A + $\mathcal{L}_{drloc}$ | 84.02 (+1.26) |
| C: | A + $\mathcal{L}_{drloc}^{*}$ | 83.14 (+0.38) |
| D: | A + $\mathcal{L}_{drloc}^{ce}$ | 83.86 (+1.10) |
| E: | A + $\mathcal{L}_{drloc}^{reg}$ | 83.24 (+0.48) |
| F: | A + $\mathcal{L}_{drloc}^{all}$ | 81.88 (-0.88) |

## 2.2 Relative positional embedding

All the loss variants presented in this section have been plugged on Swin, in which relative positional embedding is used (see the main paper, Sec. 3 and Sec. 4.1). However, the results reported in Tab. 6 show that almost all of these losses can boost the accuracy of the Swin baseline. Below, we intuitively explain why the relative positional embedding is not sufficient to allow the network to solve our localization task.

The relative positional embedding (called $B$ in [8]) used in Swin, is added to the query/key product before the softmax operation (Eq. 4 in [8]). The result of this softmax is then used to weight the importance of each "value", and the new embedding representation of each query (i.e., $e_{i,j}$, in our terminology) is given by this weighted sum of values. Thus, the content of $B$ is not directly represented in $e_{i,j}$, but only used to weight the values forming $e_{i,j}$ (note that there is also a skip connection). For this reason, $B$ may be useful for the task for which it is designed, i.e., computing the importance (attention) of each key with respect to the current query. However, in order to solve our auxiliary task (i.e., to predict $t_u$ and $t_v$ in Eq. 1 in the main paper), the VT should be able to recover and extract from a given embedding pair $(e_{i,j}, e_{p,h})$ the specific offset information originally contained in $B_{(i,j),(p,h)}$ and then blended in the value weights. Probably this is a task (much) harder than exploiting appearance information contained in $(e_{i,j}, e_{p,h})$. This is somehow in line with different previous work showing the marginal importance of positional embedding in VTs. For instance, Naseer et al. [10] show that the (absolute) positional embedding used in ViT [4] is *not* necessary for the transformer to solve very challenging occlusion or patch permutation tasks, and they conclude that these tasks are solved by ViT thank to its "dynamic receptive field" (i.e., the context represented in each individual token embedding).

# 3 Experiments with a larger training budget

Although the focus of this work is on increasing the VT training efficiency in a scenario with a limited training budget, in this section we instead investigate the effect of using our auxiliary task on scenarios with a larger training budget. Specifically, we test $\mathcal{L}_{drloc}$ with a larger number of training epochs, using higher-capacity VT models and training the VTs on ImageNet-1K.

In Tab. 7 we train both Swin and T2T on ImageNet-1K following the standard protocol (e.g., 300 epochs) and using the publicly available code of each VT baseline. When we use $\mathcal{L}_{drloc}$, we get a

slight improvement with both the baselines, which shows that our loss is beneficial also with larger datasets and longer training schedules (although the margin is smaller with respect to IN-100, see Tab. 3).

Table 7: Top-1 accuracy on ImageNet-1K. (*) Results obtained in our run of the publicly available code with the default hyperparameters of each corresponding VT baseline.

|  | Model | Top-1 Acc. |
|---|---|---|
| Swin | Swin-T | 81.2 (*) |
|  | Swin-T+$\mathcal{L}_{drloc}$ | **81.33** (+0.13) |
| T2T | T2T-ViT-14 | 80.7 (*) |
|  | T2T-ViT-14+$\mathcal{L}_{drloc}$ | **80.85** (+0.15) |

In Tab. 8, we use the Infograph dataset and we train all the networks for 300 epochs. The results confirm that $\mathcal{L}_{drloc}$ can improve the final accuracy even when a longer training schedule is adopted. For instance, comparing the results of T2T in Tab. 8 with the T2T results in Tab. 4 (100 epochs), the relative margin has significantly increased (+8.06 versus +2.62).

Table 8: Infograph, training from scratch with 300 epochs.

| Model | Top-1 Acc. |
|---|---|
| CvT-13 | 29.76 |
| CvT-13 + $\mathcal{L}_{drloc}$ | **30.31** (+0.55) |
| Swin-T | 17.17 |
| Swin-T + $\mathcal{L}_{drloc}$ | **20.72** (+3.55) |
| T2T-ViT-14 | 12.62 |
| T2T-ViT-14 + $\mathcal{L}_{drloc}$ | **20.68** (+8.06) |
| ResNet-50 | 29.34 |
| ResNet-50 + $\mathcal{L}_{drloc}$ | **30.00** (+0.66) |

Finally, in Tab. 9, we use three datasets and we train from scratch ViT-B/16 [4], which has 86.4 million parameters (about $4\times$ the number of parameters of the other tested VTs and ResNets). Note that "16" in ViT-B/16 stands for $16 \times 16$ resolution patches, used as input without patch overlapping. For a fair comparison, we used for ViT-B/16 the same image resolution ($224 \times 224$) adopted for all the other VTs (see Sec. 6), thus we get a final ViT-B/16 embedding grid of $14 \times 14$, which is pooled to get our $7 \times 7$ grid as explained in the main paper (Sec. 3). For ViT-B/16, we use $\lambda = 0.01$. Tab. 9 shows that our loss is effective also with VT models bigger than the three baselines used in the rest of the paper.

Table 9: Training from scratch ViT-B/16 with 100 epochs.

| Model | CIFAR-10 | CIFAR-100 | Infograph |
|---|---|---|---|
| ViT-B/16 | 71.70 | 59.67 | 11.79 |
| ViT-B/16 + $\mathcal{L}_{drloc}$ | **73.91** (+2.21) | **61.42** (+1.75) | **12.22** (+0.43) |

## 4 Transfer to object detection and image segmentation tasks

In this section, we provide additional fine-tuning experiments using tasks different from classification (i.e., object detection, instance segmentation and semantic segmentation). Moreover, we use a different training protocol from the one used in the main paper (Sec. 5.3). Specifically, the fine-tuning stage is standard (*without our loss*), while in the pre-training stage we either use the standard cross-entropy (only), or we pre-train the VT jointly using the cross-entropy and $\mathcal{L}_{drloc}$. We adopt the framework proposed in [8], where a pre-trained Swin VT is used as the backbone for detection and segmentation tasks. In fact, note that Swin is based on a hierarchy of embedding grids, which can be

used by the specific object detection/image segmentation architectures as they were convolutional feature maps [8].

The pre-training dataset is either ImageNet-1K or IN-100, and in both cases we pre-train Swin using 300 epochs. Hence, in case of ImageNet-1K pre-training, the baseline model is fine-tuned starting from the Swin-T model corresponding to Tab. 7 (final accuracy : 81.2), while Swin-T + $\mathcal{L}_{drloc}$ refers to the model trained with our loss in the same table (final accuracy: 81.33). Similarly, in case of IN-100 pre-training, the baseline model is fine-tuned starting from the Swin-T model corresponding to Tab. 3 (final accuracy : 89.68), while Swin-T + $\mathcal{L}_{drloc}$ refers to the model trained with our loss in the same table (final accuracy: 90.32).

The goal of these experiments is to show that the image representation obtained using $\mathcal{L}_{drloc}$ for pre-training, can be usefully transferred to other tasks without modifying the task-specific architecture or the fine-tuning protocol.

## 4.1 Object detection and instance segmentation

**Setup.** We strictly follow the experimental settings used in Swin [8]. Specifically, we use COCO 2017 [7], which contains 118K training, 5K validation and 20K test-dev images. We use two popular object detection architectures: Cascade Mask R-CNN [1] and Mask R-CNN [6], in which the backbone is replaced with the pre-trained Swin model. Moreover, we use the standard mmcv [3] framework to train and evaluate the models. We adopt multi-scale training [2, 11] (i.e., we resize the input image such that the shortest side is between 480 and 800 pixels, while the longest side is at most 1333 pixels), the AdamW [9] optimizer (initial learning rate 0.0001, weight decay 0.05, and batch size 16), and a 1x schedule (12 epochs with the learning rate decayed by 0.1 at epochs 8 and 11).

**Results.** Tab. 10 shows that Swin-T, pre-trained on ImageNet-1K with our $\mathcal{L}_{drloc}$ loss, achieves both a higher detection and a higher instance segmentation accuracy with respect to the baselines. Specifically, with both Mask RCNN and Cascade Mask RCNN, our pre-trained model outperforms the baselines with respect to nearly all detection/segmentation metrics. When pre-training with a smaller dataset (IN-100), the relative improvement is even higher (Tab. 11).

Table 10: *ImageNet-1K* pre-training. Results on the COCO object detection and instance segmentation tasks. $AP^{box}_x$ and $AP^{mask}_x$ are the standard object detection and segmentation Average Precision metrics, respectively [7].

| Architecture | Pre-trained backbone | $AP^{box}$ | $AP^{box}_{50}$ | $AP^{box}_{75}$ | $AP^{mask}$ | $AP^{mask}_{50}$ | $AP^{mask}_{75}$ |
|---|---|---|---|---|---|---|---|
| Mask RCNN | Swin-T | 43.4 | 66.2 | 47.4 | 39.6 | 63.0 | **42.6** |
| | Swin-T + $\mathcal{L}_{drloc}$ | **43.8** | **66.5** | **48.0** | **39.7** | **63.1** | 42.5 |
| | | (+0.4) | (+0.3) | (+0.6) | (+0.1) | (+0.1) | (-0.1) |
| Cascade Mask RCNN | Swin-T | 48.0 | 67.1 | 51.7 | 41.5 | 64.3 | **44.8** |
| | Swin-T + $\mathcal{L}_{drloc}$ | **48.2** | **67.4** | **52.1** | **41.7** | **64.7** | **44.8** |
| | | (+0.2) | (+0.3) | (+0.4) | (+0.2) | (+0.4) | (+0.0) |

Table 11: *IN-100* pre-training. Results on the COCO object detection and instance segmentation tasks.

| Architecture | Pre-trained backbone | $AP^{box}$ | $AP^{box}_{50}$ | $AP^{box}_{75}$ | $AP^{mask}$ | $AP^{mask}_{50}$ | $AP^{mask}_{75}$ |
|---|---|---|---|---|---|---|---|
| Mask RCNN | Swin-T | 41.8 | 60.3 | 45.1 | 36.7 | 57.4 | 39.4 |
| | Swin-T + $\mathcal{L}_{drloc}$ | **42.7** | **61.3** | **45.9** | **37.2** | **58.4** | **40.0** |
| | | (+0.9) | (+1.0) | (+0.8) | (+1.0) | (+1.0) | (+0.6) |
| Cascade Mask RCNN | Swin-T | 36.0 | 58.2 | 38.6 | 33.8 | 55.2 | 35.9 |
| | Swin-T + $\mathcal{L}_{drloc}$ | **37.2** | **59.4** | **40.3** | **34.5** | **56.2** | **36.6** |
| | | (+1.2) | (+1.2) | (+1.7) | (+0.7) | (+1.0) | (+0.7) |

## 4.2 Semantic segmentation

**Setup.** We again follow the experimental settings adopted in Swin [8]. Specifically, for the semantic segmentation experiments, we use the ADE20K dataset [15], which is composed of 150 semantic categories, and contains 20K training, 2K validation and 3K testing images. Following [8], we use the popular UperNet [13] architecture with a Swin backbone pre-trained either on ImageNet-1K or on IN-100 (see above). We use the implementation released by mmcv [3] to train and evaluate all the models.

When fine-tuning, we used the AdamW [9] optimizer with an initial learning rate of $6 \times 10^{-5}$, a weight decay of 0.01, a scheduler with linear learning-rate decay, and a linear warmup of 1,500 iterations. We fine-tuned all the models on 8 Nvidia V100 32GB GPUs with 2 images per GPU for 160K iterations. We adopt the default data augmentation techniques used for segmentation, namely random horizontal flipping, random re-scaling with a [0.5, 2.0] ratio range and random photometric distortion. We use stochastic depth with ratio 0.2 for all the models, which are trained with an input of 512×512 pixels. At inference time, we use a multi-scale testing, with image resolutions which are $\{0.5, 0.75, 1.0, 1.25, 1.5, 1.75\} \times$ of the training resolution.

**Results.** The results reported in Tab. 12 show that the models pre-trained on ImageNet-1K with the proposed loss *always* outperform the baselines with respect to all the segmentation metrics. Similarly to Sec. 4.1, when a smaller dataset is used for pre-training (IN-100), the observed relative boost is even higher (Tab. 13).

Table 12: *ImageNet-1K* pre-training. Semantic segmentation on the ADE20K dataset (testing on the validation set). mIoU and mAcc refer to the mean Intersection over Union and the mean class Accuracy, respectively. The base architecture is UperNet [13].

| Pre-trained backbone | mIoU | mAcc |
|---|---|---|
| Swin-T | 43.87 | 55.22 |
| Swin-T + $\mathcal{L}_{drloc}$ | **44.33** (+0.46) | **55.74** (+0.52) |

Table 13: *IN-100* pre-training. Semantic segmentation on the ADE20K dataset (testing on the validation set) with a UperNet architecture [13].

| Pre-trained backbone | mIoU | mAcc |
|---|---|---|
| Swin-T | 36.93 | 47.76 |
| Swin-T + $\mathcal{L}_{drloc}$ | **37.83** (+0.90) | **48.69** (+0.93) |

## 5 Training efficiency

In Fig. 3 we show the training curves corresponding to the top-1 accuracy of CvT, Swin and T2T, trained from scratch on CIFAR-100, with or without our loss. These graphs show that our auxiliary task is beneficial over the whole training stage, and it can speed-up the overall training. For instance, in case of Swin, after 60 training epochs, or method is already significantly better than the baseline full-trained with 100 epochs (55.01 versus 53.28).

Finally, we compute the overhead of $L_{drloc}$ at training time. The results reported in Tab. 14 refer to seconds per batch (with a batch size equal to 1024), and show that, overall, the overhead due to our auxiliary task is negligible with respect to the whole training time.

## 6 Implementation details and an additional ablation study on the localization MLP

Our localization MLP ($f$) is a simple feed-forward network composed of three fully connected layers. The first layer projects the concatenation of the two input token embeddings $\mathbf{e}_{i,j}$ and $\mathbf{e}_{p,h}$ into a

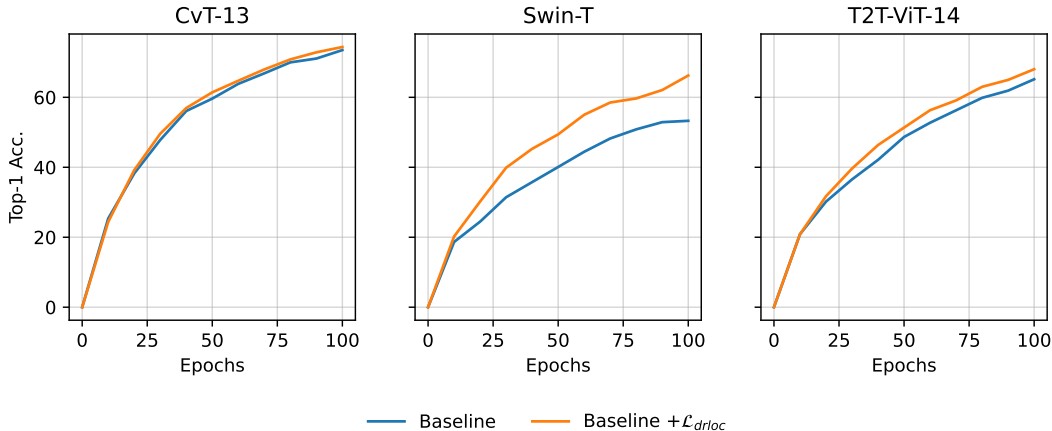

Figure 3: CIFAR-100, training from scratch, top-1 accuracy measured every 10 epochs.

Table 14: Training time comparison on CIFAR-100. The values are averaged over all training batches and jointly reported the corresponding standard deviations.

| Model | Seconds per batch |
|---|---|
| CvT-13 | $0.6037 \pm 0.0040$ |
| CvT-13 + $\mathcal{L}_{drloc}$ | $0.6184 \pm 0.0070$ (+2.43%) |
| Swin-T | $0.6684 \pm 0.0031$ |
| Swin-T + $\mathcal{L}_{drloc}$ | $0.6842 \pm 0.0033$ (+2.36%) |
| T2T-ViT-14 | $0.5941 \pm 0.0053$ |
| T2T-ViT-14 + $\mathcal{L}_{drloc}$ | $0.6046 \pm 0.0058$ (+1.77%) |

512-dimensional vector and then it applies a `Relu` activation. Next, we use a linear layer of dimension 512 followed by a `Relu` activation. Finally, we use a linear layer dedicated to the prediction, which depends on the specific loss variant, see Sec. 2. For instance, in $\mathcal{L}_{drloc}$, the last layer is composed of two neurons which predict $d_u$ and $d_v$. The details of the MLP head are shown in Tab. 15, while in Tab. 16 we show the influence of the number of neurons in the hidden layers of $f$.

Table 15: The details of the localization MLP head. $d$ is the dimension of a token embedding. The number of outputs $o$ and the final nonlinearity (if used) depend on the specific loss. In $\mathcal{L}_{drloc}$, $\mathcal{L}_{drloc}^*$ and $\mathcal{L}_{drloc}^{all}$, we use $o = 2$ without any nonlinearity. Converesely, in both $\mathcal{L}_{drloc}^{ce}$ and $\mathcal{L}_{drloc}^{reg}$, the last layer is split in two branches of $2k + 1$ neurons each, and, on each branch, we separately apply a `SoftMax` layer.

| Layer | Activation | Output dimension |
|---|---|---|
| Input | - | $d * 2$ |
| Linear | ReLU | 512 |
| Linear | ReLU | 512 |
| Linear | - / SoftMax | $o$ |

In our experiments, we used the officially released framework of Swin [8][2], which also provides all the necessary code to train and test VT networks (including the object detection and segmentation tasks of Sec. 4). For a fair comparison, we use the official code of T2T-ViT [14][3] and a publicly released code of CvT [12][4] and we insert them in the training framework released by the authors

---

[2]https://github.com/microsoft/Swin-Transformer
[3]https://github.com/yitu-opensource/T2T-ViT
[4]https://github.com/lucidrains/vit-pytorch

Table 16: CIFAR-100, 100 epochs, training from scratch: the influence of the number of neurons used in each of the two hidden layers of the localization MLP.

| Model | Number of neurons | | |
|---|---|---|---|
| | 256 | 512 | 1024 |
| CvT-13 + $\mathcal{L}_{drloc}$ | 74.19 | **74.51** | 73.80 |
| Swin-T + $\mathcal{L}_{drloc}$ | 65.06 | **66.23** | 64.33 |
| T2T-ViT-14 + $\mathcal{L}_{drloc}$ | 66.49 | **68.03** | 67.83 |

of Swin. At submission time of this paper, the official code of CvT [12] was not publicly available. Finally, the ViT-B/16 model used in Sec. 3 is based on a public code[4].

When we train the networks from scratch (100 epochs), we use the AdamW [9] optimizer with a cosine decay learning-rate scheduler and 20 epochs of linear warm-up. We use a batch size of 1024, an initial learning rate of 0.001, and a weight decay of 0.05. When we fine-tune the networks (100 epochs), we use the AdamW [9] optimizer with a cosine decay learning-rate scheduler and 10 epochs of linear warm-up. We use a batch size of 1024, an initial learning rate of 0.0005, and a weight decay of 0.05. In all the experiments, the images of all the datasets are resized to the same fixed resolution $(224 \times 224)$.