# OpenReview forum: "Efficient Training of Visual Transformers with Small Datasets"
_NeurIPS.cc/2021/Conference — NeurIPS 2021 Poster_

### Official Review · Reviewer_Tk2H · 2021-07-04

**Rating:** 5
**Confidence:** 4

**Summary:**

The authors compare the robustness of three existing visual transformers (VTs) in a small training-set regime and propose a self-supervised pretext task with a corresponding loss function to extract additional information from the images. Specifically, the authors find that VTs are more data-hungry than common CNNs due to the lack of inductive bias and different VTs have largely different behaviors when training data are scarce. To solve this, the authors propose to learn spatial relations within an image by predicting the relative positions of pairs of tokens extracted from the same image. Experimental results on 13 datasets demonstrate the effectiveness of the proposed method. However, some details are unclear and the experiments can be further improved. Please see my comments below for details.

**Limitations And Societal Impact:**

Yes

**Main Review:**

**Contributions**:
1.	The authors quantitatively show that different VTs have largely different performances when trained from scratch with small datasets or few training epochs.

2.	This paper presents a new self-supervised relative localization loss and multiple variants, which is able to increase the generalization ability of trained VTs.

3.	Experimental results show that the proposed localization task is able to improve the performance and training speed of the VTs on small datasets.

**Questions and points needed to be improved**:
1.	Some important detail is missing. In Eq. (6), what does $\alpha$ denote?

2.	SAM [1] also focuses on improving the generalization of VTs. It would be better for the authors to provide more discussions on the difference between the proposed method and SAM.

[1] Sharpness-aware Minimization for Efficiently Improving Generalization, ICLR, 2021.

3.	The coefficient $\lambda$ of relative localization loss in the final loss of three VTs is different. How to determine these hyper-parameters? More experiments or explanations are required.

4.	The authors state that the proposed method focuses on small datasets. However, the generalization of the proposed method on large-scale datasets is also important. It would be better for the authors to provide more results on the large-scale dataset.

5.	The authors state that the bad performance of the last loss variant is due to the finer resolution grids of intermediate blocks in Swin. Can we use the pooling operation to solve this issue?

6.	The authors propose to impose a Gaussian prior on $P_u$ and $P_v$ as the third variant to alleviate the problem in the second loss variant that prediction errors are independent of the “distance”. However, in Tab. 1(b), the performance of the proposed method with a Gaussian prior (row E) is worse than that without a Gaussian prior (row D). Please give more explanations.

7.	In Tab. 2(a), it is unclear why the performance of Swin-T goes worse with the increase of $m$? Please give more analyses.

8.	Experiments are not sufficient. The results in the experiment can not show that the proposed method is able to speed up the training. To demonstrate this, it would be better for the authors to show the training curve of different methods in terms of the loss or classification accuracy.

9.	There are some typos in the paper.
(1)	In Eq. (5), the closing parenthesis of the log function is missing.
(2)	In Line 250 of Page 6, “… imposes …” should be “… to impose …”.
(3)	In Line 272 of Page 7, “… mekes …” should be “… makes …”.


**Time Spent Reviewing:**

20

---

> ### Author Response · Authors · 2021-08-10
> **Answers to Reviewer Tk2H**
>
> **1.Q. In Eq. (6), what does α denote? A.** α weights the importance of the Gaussian prior [17]. We omitted this detail for brevity, but we will clarify it in the revised version.
>
> **2.Q. The difference between the proposed method and SAM. A.** SAM is a generic optimization method, which is based on the connection between the sharpness of the loss landscape and the generalization capabilities of a neural network. It is based on an optimization technique in which the loss value is minimized jointly with an estimate of the loss sharpness. In contrast, our goal is to extract additional information from images without the need of additional annotation. Specifically, we exploit the geometric grid of the final VT token embeddings to predict their pairwise distances, this way encouraging the VT to represent location-specific information. We will include this comparison in Sec. 2.
>
> **3.Q. Ablation on the value of lambda. A.** You are correct, we agree on the importance of showing an empirical ablation on the lambda value. We report below the analysis we performed in the beginning of our work (using CIFAR-100), which motivates our choice of the lambda values as reported in l. 214-215 (we will include this table in the final version).
>
> |Model|0.1|0.5|1.0|
> |:----|:----:|:----:|:----:|
> |CvT-13|68.03|67.03|66.53|
> |Swin-T|58.15|66.23|64.28|
> |T2T-ViT-13|74.74|74.07|72.84|
>
> Table: Ablation study on lambda.
>
> **4.Q. Large-scale dataset experiments. A.** Please, see our Answer 2 to Reviewer ndSd.
>
> **5.Q. Can we use the pooling operation in the finer resolution grids of the intermediate blocks in Swin for the last loss? A.** This an interesting idea, thank you for your suggestion. We did not try that, but indeed a pooling layer may be applied to each block before feeding the embeddings to the (block-specific) localization MLP, similarly to what we do for the last grid of T2T and CvT. Since this pooling operation would be used only for the localization MLP, it would not influence or decrease the representation capacity of the native VT.
>
> **6.Q. The Gaussian-prior based loss underperforms the cross-entropy loss. A.** The loss introduced in Eq. 6 is inspired by [17]. The reason why its results are worse than the cross-entropy loss may be due to the lack of tuning of the parameter alpha (see above), for which we used the values suggested in [17]. However, we believe that a simpler loss (e.g., the one presented in Eq. 2 and used in all the other experiments) is easier to be reproduced, having less hyper-parameters to tune.
>
> **7.Q. Tab. 2 (a), it is unclear why the performance of Swin-T goes worse with the increase of m? A.** We presume this is due to the fact that, with too many sample pairs per batch, the localization MLP starts to overfit on our pretext task.
>
> **8.Q. Training curve of different methods. A.** Following your suggestion, we report below the top-1 accuracy measured every 10 epochs on CIFAR-100:
>
> - CvT-13:
>   - original: [25.35, 38.23, 47.81, 56.09, 59.63, 63.85, 66.86, 69.99, 71.08, 73.50]
>   - +our loss: [24.38, 39.18, 49.59, 56.94, 61.42, 64.74, 67.97, 70.83, 72.86, 74.38]
>
> - Swin-T:
>   - original: [18.65, 24.46, 31.43, 35.76, 40.08, 44.43, 48.23, 50.84, 52.90, 53.28]
>   - +our loss: [20.21, 30.21, 39.88, 45.28, 49.41, 55.01, 58.51, 59.68, 62.08, 66.23]
>
> - T2T-ViT-14:
>   - original: [20.87, 30.18, 36.53, 42.12, 48.66, 52.73, 56.29, 59.85, 61.91, 65.16]
>   - +our loss: [20.86, 31.72, 39.63, 46.36, 51.34, 56.31, 59.12, 63.00, 64.98, 68.03]
>
> We will plot these training curves in the Supplementary Material.
>
> **9.Q. There are some typos. A.** We will carefully proofread the whole paper, thank you for your suggestions.

---

> > ### Comment · Reviewer_Tk2H · 2021-09-02
> > **Feedback on rebuttal**
> >
> > I have read all the comments and thank the authors for their replies. However, my concern about the Gaussian-prior based loss remains and I follow the opinions of other reviewers that the paper misses detailed analysis on the performance of those loss variants which require further discussions. I tend to keep my score.

---

> > > ### Author Response · Authors · 2021-09-02
> > > **Re: Feedback on rebuttal**
> > >
> > > Thank you very much for your answer. We regret to see that your opinion on our paper strictly depends on the loss variants, because this is *not* the main part of our work. Indeed, in the revised version of the paper, following the suggestion of Reviewer 44YH, we will move most of the discussion about these variants to the Supplementary Material. The main part of our paper is the use of a self-supervised pretext task, based on the relative displacement of token pairs (densely sampled), which is used as an auxiliary task to regularize VT training. The experiments which confirm the soundness of our proposal are shown in Sec. 5.2 and 5.3.
> > >
> > > On the other hand, the loss variants (which implement the pretext task in slightly different ways) are empirically analysed in Sec. 5.1 (and in the Supplementary Material, Sec. 3). In the revised version of the paper, we will provide a deeper analysis of these variants (in the Supplementary Material), but we emphasize once more that this is *not* the core of our proposal. Specifically, following your suggestion, we run an additional experiment to compare the Gaussian-prior based variant with the cross-entropy and the L1 loss. This experiment follows the same setup of Tab. 1 (b), but it includes the other two baselines (CvT and T2T). The results are shown in the table below.
> > >
> > > |Model|Original|+L^{reg}_{drloc}|+ L^{ce}_{drloc}|+ L_{drloc}|
> > > |:----|:----:|:----:|:----:|:----:|
> > > |CvT-13|85.62|85.78|85.62|**85.98**|
> > > |Swin-T|82.76|83.24|83.86|**84.02**|
> > > |T2T-ViT-14|82.74|83.26|83.38|**83.90**|
> > >
> > > *Table: IN-100, 100 epoch training, a comparison between 3 different loss variants using all the VT baselines. L^{reg}_{drloc} is based on the default alpha value reported in [17].*
> > >
> > >
> > > The relative ranking of these 3 loss variants is the same both when using Swin and when using T2T. When using CvT, the Gaussian-prior loss slightly outperforms the cross-entropy loss, but the L1 loss is still the best over the 3 variants.
> > >
> > > Finally, please note that, as mentioned in our previous answer, most likely a higher accuracy of the Gaussian-prior variant can be obtained by tuning its specific alpha hyperparameter (remember we used the default value suggested in [17]). We are currently running a grid search on this alpha value, and we will post the results of this experiment if it will be ready by the end of the discussion period. However, one of the goals of our work is to provide an *easy-to-use* regularization method for VT training. Thus, the necessity to tune an additional parameter (alpha) makes this loss variant less useful than the others, and this is the reason why we chose the default alpha value proposed in [17]. In other words: even if there is an alpha value which makes the Gaussian-prior loss more effective than the other losses (thus, possibly *improving* the results of our method presented in Sec. 5.2 and 5.3), this comes at the cost of having to tune an additional hyperparameter (please, see also Answer 1 to Reviewer jUVG).

---

> > > > ### Author Response · Authors · 2021-09-03
> > > > **Re: Feedback on rebuttal - 2**
> > > >
> > > > As mentioned in our previous comment, and following your suggestions, we tuned the alpha hyperparameter of the Gaussian prior loss. As shown in the following table, it seems that the default value (0.001), recommended in [17], is the best option.
> > > >
> > > > |0.01|0.001(default)|0.0001|
> > > > |:-----:|:-----:|:-----:|
> > > > |83.18|83.24|83.04|
> > > >
> > > > *Table: Swin-T +L^{reg}_{drloc}, trained on IN-100 with different alpha values.*
> > > >
> > > > Both this experiment and the previous one show that the L1 loss variant is not only the easiest-to-use (less hyperparameters)  but it also achieves the best performance.

---

### Official Review · Reviewer_8uyZ · 2021-07-12

**Rating:** 6
**Confidence:** 3

**Summary:**

This paper presents:
- a comparison of some recent transformer architectures (CvT, Swin and T2T) on small(er) datasets with limited computational ressources
- a simple self-supervised loss (predicting relative position with L1 loss) that can be directly added to regularize and improve performances.
It includes results on 5 standard datasets, and a study of several relevant alternative for the self-supervised loss (using the same pretext task)

**Ethical Concerns:**

I see no ethical concerns

**Ethics Review Area:**

["I don’t know"]

**Limitations And Societal Impact:**

I think some limitations and results compared to SoA should be discussed in more details (see above)
I see no societal impact issue.

**Main Review:**

Not being an expert in recent vision transformers, I enjoyed reading this paper and felt that despite it's very limited novelty it was useful. It was in any case well written and easy to read, with meaningful comparison and many experiments. Thus, despite some limitations that need to be addressed (most importantly 1 and 2 bellow) I fell the paper can be accepted

limitations:
1. I would like to know how much the proposed training have converged: Table 2 hints 100 epochs (used in all the rest of the paper) is really far from convergence, and the results might not hold after convergence. I feel this is a bit hidden/not outlined enough in the paper, and I think a clear analysis on this is necessary
2. Since the paper is using standard dataset, I would like to see a clear comparison (e.g. in Tab 3) with State of the Art and standard (ResNet) results on these datasets. It seems to me the reported results are quite far (which is ok but should be clearly stated and visible in the tables)
3. I would have liked to see a detailed study of the influence of the size of the localization MLP: as hinted by the authors, it is probably crucial to the quality of the results
4. I am not sure some of the results are significative, in particular:
* the boost of the selected L1 loss over over the CE alternative loss in Table 1. I would actually like to see this table for all three architectures.
* the boost after 300 training epochs for CvT in Table 2
5. I am missing an intuition on why an absolute value in the relative localization helps

misc., l. 114, grammar issue

**Time Spent Reviewing:**

1.5

---

> ### Author Response · Authors · 2021-08-10
> **Answers to Reviewer 8uyZ**
>
> **1.Q. Training till convergence. A.** Estimating the number of epochs necessary for a model to converge on a given dataset/task is not easy. Previous work has used 300 epochs for training VTs on ImageNet [59, 34, 56, 58, 57, 33]. Please, see Answer 2 to Reviewer ndSd for our new ImageNet-1k experiments with 300 epochs.  Concerning ImageNet-100 (Tab. 2 (a)) you are correct: the improvement of our loss w.r.t. the standard training for all the 3 tested VTs is smaller with 300 epochs than when using 100 epochs. However, this is somehow expected, being our method a regularization approach, which typically help more with a smaller training budget (see l. 301-307). Finally, concerning the other datasets, we used 100 epochs because, as a common practice, smaller datasets are trained with less epochs to avoid overfitting (and also because we wanted to investigate a scenario with a limited training budged, l. 301-302). However, following your suggestion, we used Infograph and we trained (from scratch) all the models for 300 epochs. The table below shows that the improvements observed with 100 epochs are persistent even with a longer training. For Swin and T2T, the gain is even larger (in the latter case, much larger: more than 8 points vs. 2.62 points obtained with 100 epochs).
>
> |Method| Acc@1|
> |:----|:----:|
> |CvT-13|29.76|
> |CvT-13 + L_{drloc}|30.31|
> |Swin-T|17.17|
> |Swin-T + L_{drloc}|20.72|
> |T2T-ViT-14|12.62|
> |T2T-ViT-14 + L_{drloc}|20.68|
>
> Table: Infograph with 300 training epochs.
>
> **2.Q. Comparison with standard (ResNet) results on these datasets. A.** Following your suggestion, we used CIFAR-10, CIFAR-100 and Infograph and a standard ResNet-50 and we repeated the experiments of Tab. 3 with this CNN. The table below shows the results. In the second row, we also report the results obtained using the ResNet baseline jointly with our pretext task. In more detail, we replace the VT token embedding Grid (G_x in Eq. 2) with the last convolutional layer of the ResNet and we apply our loss (Eq. 2) on this convolutional feature map. The results of the table show that, on the 3 tested datasets: (1) a standard ResNet-50 gets results similar to CvT (but much higher than the other two VT baselines); (2) **Our loss is useful also when used with a ResNet.**
>
> ||CIFAR-10|CIFAR-100|Infograph|
> |:----|:----:|:----:|:----:|
> |ResNet-50|91.78|72.80|19.81|
> |ResNet-50+L_{drloc}|92.03|72.94|20.79|
>
> Table: ResNet-50 results, without and with our loss.
>
> In the revised version, we will add the results for all the other datasets (experiments not yet finished).
>
> **3.Q. The influence of the size of the localization MLP. A.** We report below an analysis of the influence of both the number of layers and the number of neurons in the hidden layers of the localization MLP. In the former case, we keep fixed to 512 the number of hidden layer neurons. Please, note that there is a typo on line 208, since in all the experiments of our paper we used a localization MLP with 2 hidden layers (hence, we have a total number of 3 layers, as correctly reported in the Supplementary Material, l. 65).
>
> |Model|1|2|3|4|
> |:----|:----:|:----:|:----:|:----:|
> |CvT-13|74.07|74.19|74.51|74.69|
> |Swin-T|59.85|63.27|66.23|66.64|
> |T2T-ViT-14|66.32|66.75|68.03|66.55|
>
> Table: number of layers in the MLP head.
>
>
> |Model|256|512|1024|
> |:----|:----:|:----:|:----:|
> |CvT-13|74.19|74.51|73.80|
> |Swin-T|65.06|66.23|64.33|
> |T2T-ViT-14|66.49|68.03|67.83|
>
> Table: number of neurons in the hidden layers of the MLP head (3 layers).
>
> Despite slightly better results can be obtained with a deeper MLP in case of CvT and Swin, for simplicity we prefer to uniformly use 3 layers with all the 3 VT baselines.
>
> **4.Q. Comparing the L1 loss with the CE loss in Table 1 using all three architectures. A.** Following your suggestion, we compared the two losses using all the 3 VT baselines and ImageNet-100. The table below shows that L1 (slightly) outperforms CE with all the baselines.
>
> |Model|Original|+ L^{ce}_{drloc}|+ L_{drloc}|
> |:----|:----:|:----:|:----:|
> |CvT-13|85.62|85.62|85.98|
> |Swin-T|82.76|83.86|84.02|
> |T2T-ViT-14|82.74|83.38|83.90|
>
> Table: Ablation study on the two loss variants on ImageNet-100.
>
> **5.Q. Why an absolute value in the relative localization helps. A.** We presume this is due to the fact that the task is simplified.

---

### Official Review · Reviewer_jUVG · 2021-07-14

**Rating:** 7
**Confidence:** 5

**Summary:**

The paper tackles the issue of low performance of recently proposed Vision Transformers in small data regime, by adding an extra self-supervision loss in addition to classification loss during training. The proposed loss regresses offsets between randomly selected embeddings, and is supposed to act as a regularizer. The approach is validated experimentally on a number of datasets, with an ablation study on the type of self-supervision loss and the number of training epochs. The approach provides a significant boost to ViTs of various architectures in small data regime evaluations.

**Limitations And Societal Impact:**

The paper uses a common image classification setting, I do not see potential negative societal impacts from their work.

**Main Review:**

The paper tackles the issue of low performance of recently proposed Vision Transformers in small data regime. Improving the performance of Vision Transformers in this setting would enable researchers with smaller computational budgets to experiment with Transformers and  potentially faster research iteration speed. Additionally, applying Transformers in setting where fine-tuning is not possible and data is scarce, yet global context reasoning of Transformers is desired, might be unlocked.

The authors propose to add self-supervision task in addition to main classification head of ViT. The self-supervision task is to regress relative offsets between pairs of chosen random number of embeddings. They experimentally show that the proposed extra training loss improves performance of various recent ViT architectures on a number of small datasets. They also ablate the choice of the extra training loss (regression, classification, regression with dense prior).

Pros:
- the motivation behind adding the extra self-supervision loss is clear, and has been explored in other Transformer applications before
- the paper is easy to follow and clearly written
- the approach is simple and should be straightforward to implement, and also has minimal computational cost

Cons:
- the approach adds multiple hyperparameters (number of loss locations, weighting of the loss wrt the classification loss) to training
- the experimental evaluation lacks analysis of statistical significance of the results. In small data regime the final classification accuracy often has high variance, so it is possible that conclusions drawn on a single experiment might not actually support experiment hypothesis. In particular, in table 1b the authors conclude that the best choice for their supervision loss type based on a single run for each type, and used it for all experiments. Moreover, test set is used instead of a separate validation set for the type tuning. I would suggest to create a separate validation set, and train multiple models with different random seed, or to do e.g. 5-fold validation, and report mean and std on the validation set for each loss type, to reinforce the hypothesis.

**Time Spent Reviewing:**

3

---

> ### Author Response · Authors · 2021-08-10
> **Answers to Reviewer jUVG**
>
> **1. Q. Multiple hyper-parameters (number of loss locations, weighting of the loss wrt the classification loss). A.** We indeed have 2 hyper-parameters: “m” (the number of embedding pairs) and “lambda”, the relative weight of our loss. However, it is very hard to develop a new approach without new hyper-parameters, and, overall, the number of hyper-parameters of the native VT baselines are many more than the 2 new ones we add. Importantly, we have kept fixed these hyper-parameter values (apart from the VT-specific value of lambda) across all the experiments, datasets, main tasks, etc., and this shows that adopting the values we suggest in our paper is a robust choice, without the need to heavily tune our specific hyper-parameters for each possible application.
>
> **2. Q. Statistical significance of the results. A.** Please, see Answer 3 to Reviewer 44YH.
>
> **3. Q. 5-fold validation for Tab. 1 (b). A.** We agree that a 5-fold validation is more correct. Unfortunately, we did not have enough time to run this experiment, since, in this short rebuttal time, our computational resources have been busy with (many) other experiments asked by the Reviewers. We will use a 5-fold validation for Tab. 1 (b) as you suggested in the final version. However, please, refer also to Answer 4 to Reviewer 8uyZ, in which we performed a similar experiment.

---

### Official Review · Reviewer_ndSd · 2021-07-17

**Rating:** 4
**Confidence:** 4

**Summary:**

In this paper, the authors propose a self-supervised loss to improve ViT-based models' accuracy on small datasets. The motivation is that ViTs are more flexible than CNNs, and thus overfit more on small datasets; The authors thus aim to provide additional supervision to help these ViTs to work better. Concretely, what the authors study is a relative localization task that trains the network to predict the relative spatial relationship between two feature vectors located at different spatial locations. On multiple small datasets, the authors demonstrate good improvement over the original networks that don't use the self-supervised loss.


**Limitations And Societal Impact:**

My main concerns are regarding the lack of comparison to prior work with the proposed self-supervised task and the usefulness of the small-dataset trained-from-scratch settings.

As of societal impact, I don't see anything particularly concerning.

**Main Review:**


**[originality: medium]**

While the high-level idea of using self-supervised loss to improvement performance when supervision is scare is not new (e.g., semi-supervised learning has a related idea), the relative location task is interesting and novel.

**[quality: low-medium]**

 - I find the ablation study limited. For example, there have been many other self-supervised tasks proposed (e.g., [15, 41, 42, 38] to name a few), but the authors didn't compare to any of them. It's unclear what the pros and cons of the proposed task is.
 - While this paper focuses on small dataset, how the effectiveness of the proposed method changes with dataset size is also an important thing to understand (i.e., when does the proposed method work?).

**[clarity: medium-high]**

I found the paper clearly written and easy to follow. The method and implementation details are described with a satisfactory level of details.

**[significance: low]**

The main concerns I have are as follows
 - The main message from the experiments presented is that using the proposed additional self-supervision helps on small datasets. I find this expected and hard to draw new knowledge from. (For example, in the extreme case of having only 1 example in a dataset, the supervised learning will fail completely, while adding self-supervised training signals is expected to help.)
 - The authors argue that ViTs require very large datasets to train. This is not entirely true; prior work (e.g., [51]) shows that on the small dataset of ImageNet-1K, ViTs can be successfully trained with proper training procedures. I wonder why the authors not include ImageNet-1K, but only even smaller datasets which typically are not trained from scratch.

**Time Spent Reviewing:**

1.5 hours

---

> ### Author Response · Authors · 2021-08-10
> **Answers to Reviewer ndSd**
>
> **1.Q. The ablation study is limited. For example, the authors didn't compare to other self-supervised tasks (e.g., [15, 41, 42, 38]). A.** This question is not very clear to us. As a common practice, in the ablation study, we have analyzed the different elements of our own method (e.g., different losses, different hyper-parameter values, etc.). Conversely, if you mean that we should compare our approach with other self-supervised methods (so, not an “ablation” study but a state-of-the-art comparison), this is a bit out of the scope of this paper. In fact, we do not propose a fully self-supervised approach, but rather a method in which a specific self-supervised pretext task is used jointly with standard supervised VT training. Hence, it is not clear how to compare other self-supervised methods with our approach, since we also use labels during training. Note also that the papers you mention (and almost all the computer vision based self-supervised papers published so far) do not use VTs but CNNs, so a comparison would be not easy even in the case our method would be fully self-supervised. Finally, if instead you mean that we should replace our translation offset prediction task with one of the pretext tasks proposed in [15, 41, 42, 38], this is an interesting idea, but it is not clear how it should be developed. For instance, the Jigsaw puzzle task [41] is based on re-ordering a random permutation of a grid of 3X3 patches (9! total possible combinations, which are subsampled in [41], to avoid having too many final linear classifiers). However, scaling this approach to a grid of 7X7 elements (49! total combinations), would lead to an intractable number of parameters in the localization MLP (please, remember that our goal is to use a task densely defined over the whole embedding grid). We leave these suggestions for future work.
>
> **2.Q. How the effectiveness of the proposed method changes with dataset size. A.** Please, note that the 11 datasets we used have a quite variable size (see Tab. 1 (a)). However, following your suggestion, we used ImageNet-1k to train both the baselines and our models for 300 epochs. We report the results in the table below, where the column “Original” corresponds to our run of the publicly available code of the VT baseline.
>
> |Model|Original|+ L_{drloc}|
> |:----|:----:|:----:|
> |Swin-T|81.2|81.33|
> |T2T-ViT-14|80.7|80.85|
>
> Table: ImageNet-1k experiments.
>
> We are currently training CvT, and we hope the results will be ready by the end of the NeurIPS discussion phase (anyway they will be reported in the final version of the paper). These ImageNet-1k results show that our loss, despite being designed to regularize small-scale datasets and few training epochs, is not decremental and it can still help to (slightly) improve the results when used with datasets of the size of ImageNet-1k.
>
> **3.Q. Additional self-supervision helps on small datasets is expected. A.** Strictly speaking, our approach is a multi-task learning method (l. 144-147). In the literature of multi-task learning, it is well-known that, adding additional tasks to the main task, it may be beneficial or harmful depending on the overall compatibility of the tasks. For instance, the phenomenon known as “negative transfer” indicates a negative contribution of the new tasks when the latter are dissimilar from the main task (independently of the size of the dataset). Thus, the fact that our pretext task is beneficial for all the datasets, all the VT baselines and all the main tasks we tested (including the object detection and the segmentation tasks in the Supplementary Material) is not a trivial finding. We believe that this is due to the fact that the proposed pretext task is general enough to help the VT to extract additional, useful information from the training samples, but we highlight once more that this result is not trivial. For instance, Tab. 1 (b), row F, shows that the last loss obtains results inferior to the baseline, which shows that, even when using the same basic pretext task, but not properly implemented, we can observe a negative transfer.
>
> **4.Q. Prior work (e.g., [51]) shows that on ImageNet-1K, ViTs can be successfully trained. A.** This is true, but [51] is based on a distillation approach, where ViT is trained using a pre-trained CNN (l. 101). This means you need to train 2 models, while our method can directly regularize a standard VT training. We refer to Answer 2 for the ImageNet-1K results.
>
> **5.Q. Usefulness of the small-dataset trained-from-scratch settings. A.** We agree that, with small datasets, fine-tuning is the most common strategy in computer vision tasks. We believe that our experiments in Tab. 4 show the effectiveness of our regularization loss in this scenario, which gives a boost in **all** the 10 tested datasets, which may be more than 5/7 points in some cases (please, note that there is a typo in Tab. 4: on ClipArt, the difference between ours and T2T is +3.63 and not +0.37). However, we also believe that training-from-scratch is important in some scenarios in which the domain shift w.r.t. ImageNet is too large for the fine-tuning procedure to be effective (l. 317-322). For instance, if the VT architecture should be changed to be adapted to a specific task and/or to deal with sensory data different from RGB images, then all or some of the layers need to be trained from scratch on the new domain. Other application examples regard specific domains such as medical image analysis, industrial-defect inspection, etc., where the images are not “natural” scenes as those contained in ImageNet, so pre-training may be less effective than training from scratch. Last but not least, it is worth noting that in these contexts (e.g., medical/industrial image analysis), the training datasets are usually quite small.

---

> > ### Comment · Reviewer_ndSd · 2021-08-31
> > **Re: Answers to Reviewer ndSd**
> >
> > I'd like to thank the authors for providing the author feedback.
> >
> > One simple way to compare to existing self-supervised work is through simple fine-tuning. For example, [23, 10], etc. show that SSL+fine-tuning is effective in many downstream tasks. I agree that strictly speaking, the proposed method is multi-task learning instead of pure SSL, but importantly it uses "self-supervision" to improve recognition; I thus think SSL literature should be discussed and compared in the scope of this paper.
> >
> > I agree that the "small-dataset trained-from-scratch settings" might be useful in some cases and fine-tuning might fail in some scenarios, but I also think without providing more extensive empirical comparison with fine-tuning it's hard to draw conclusions from.
> >
> > Overall, after reading all author feedback and all reviews, most of my main concerns remain, and I'd keep my recommendation unchanged.

---

> > > ### Author Response · Authors · 2021-09-01
> > > **Answers to Reviewer ndSd (new comments)**
> > >
> > > Thank you very much for your kind answer and for your constructive feedback. We answer below to all the points.
> > >
> > > **Q. Empirical comparison with previous self-supervised work. A.** Unfortunately, we cannot compare with previous Self-Supervised Learning (SSL) work simply because we use labels in our training procedure (the comparison would be in our favour and it would not be fair w.r.t. previous SSL work). For instance, the fine-tuning evaluation protocol you suggest (e.g., adopted in [23,10]), is based on 2 stages: (1) fully-unsupervised pre-training on ImageNet-1k (using a specific SSL method) and (2) standard supervised fine-tuning on a downstream task (e.g., object detection or semantic segmentation, etc.). However, unfortunately, this cannot be done in our case, being our Stage (1) non fully-unsupervised.
> > >
> > > Please, note that, in the Supplementary Material (Sec. 2), we provide additional fine-tuning experiments using object detection, instance segmentation and semantic segmentation downstream tasks. In those experiments, we use the Swin detection/segmentation code, and we compare (a) Swin pre-trained using the standard supervised cross-entropy loss with (b) Swin pre-trained using the cross-entropy loss jointly with our loss. The fine-tuning stage is standard supervised in both cases. These experiments show that the image representations obtained using our method can be transferred to downstream tasks and are better than those obtained by pre-training using standard supervision only. Note that this comparison is fair, because in Stage (1) (and in Stage (2) as well), both (a) and (b) are pre-trained using *the same amount of labelling information*.
> > >
> > > **Q. Comparison with self-supervised literature. A.** We actually compare with previous SSL literature in Sec. 2 (l. 129-163) and in Sec. 1 (l. 71-73). As far as we know, the discussion in Sec. 2 includes all the most recent important SSL works, including many arXiv papers not yet officially published.
> > >
> > > **Q. More extensive empirical comparison with fine-tuning. A.** Please, note that in our paper we provide an extensive fine-tuning comparison using 10 different datasets and 3 different VT baselines in Tab. 4., and, as aforementioned, additional fine-tuning experiments are presented in the Supplementary Material (with 3 downstream tasks, using both the COCO and the ADE20K datasets, jointly with 3 basic detection/segmentation architectures, Mask RCNN, Cascade Mask RCNN and UperNet). We believe that this fine-tuning based evaluation is strong enough, especially because when using our loss we *always* get an improvement w.r.t. standard training, independently of the task, the dataset, the VT baseline, etc. (please, remember that there’s a typo in Tab. 4 and that our gain w.r.t. T2T on ClipArt is +3.63). Of course, other datasets, tasks or baselines can be used, but it is also important to take into account that computational resources are always limited, and so also the possible experiments are unfortunately limited... However, if you think there is a specific important fine-tuning experiment which is missing and can be tested using our framework, we would be more than happy to try it.
> > >
> > > Finally, for completeness, we also repeated the ResNet-50 based experiments (please, see our previous Answer n. 2 to Reviewer 8uyZ) using a fine-tuning protocol, and we report below our results (averaged over 5 different runs and including the standard deviations):
> > >
> > > ||CIFAR-10|CIFAR-100|Infograph|
> > > |:----|:-----:|:-----:|:-----:|
> > > |ResNet-50|97.65 (+- 0.03) |85.44 (+- 0.09) |44.30 (+- 0.04) |
> > > |ResNet-50+L_{drloc}|97.74 (+- 0.05) |85.65 (+- 0.04) |44.39 (+- 0.07) |
> > >
> > > Table: ResNet-50 fine-tuning (100 epochs) based on an ImageNet-1K standard pretrained model. Fine-tuning is done with or without our loss.
> > >
> > > When using ResNets, the improvement of our loss is marginal, but it is consistent in all the 3 tested datasets. Despite the goal of our work is to propose a regularization technique for VTs, these CNN based experiments show that out pretext task is more general, and it may be useful also with ResNets.

---

### Official Review · Reviewer_44YH · 2021-08-02

**Rating:** 5
**Confidence:** 3

**Summary:**

This paper proposes a new self-supervised pretext task for vision transformer models to improve training on small-medium size datasets. The paper also conducted extensive experiments to show this method can improve the performance on various datasets with different vision transformer architectures.


**Ethical Concerns:**

There are no ethical issues with this paper as far as I concern.

**Limitations And Societal Impact:**

The paper has a section dedicated to the limitations of this paper.
There are two major contributions the paper proposes to make. With regard to the first one, more extensive experiments are needed and the result section can provide better comparisons to prove its point (e.g. quantifying the differences in accuracies among different VT architectures in the table). For the second contribution, the method itself proposed by the paper is not very different from ELECTRA but seems to work well. The paper needs to provide more convincing results to show that this method is truly effective.

**Main Review:**

This paper is clearly-written and well-organized. The method is clearly formulated in mathematical terms and its differences from previous methods are carefully explained and well-cited. The main contribution of this paper is the design of the new pretext task used as a regularizer with any kind of vision transformer models. In fact, the major difference lies in the way token embeddings are sampled. The novelty of this design is minor.

However, there are also a few aspects of this paper that might require further discussions.
Firstly, it seems that all the loss variants proposed in the paper do not outperform the “vanilla” loss version $\mathcal{L}_{drloc}$ proposed at first. Most of the experiments are also done with vanilla version except for those in the first part of the ablation studies. I think it is a bit unnecessary for the paper to spend much space talking about those variants that do not actually perform well. Besides, the paper also did not provide detailed analysis on the performance of those loss variants, making introduce loss variants at beginning less meaningful.

Secondly, here are some thoughts on the experimental design.
It is not found in the paper how many different random seeds are used to train each model. Therefore, we cannot tell if the improvement is statistically significant, especially in those cases where the improvement is only around 0.1.
Moreover, the paper uses the hyper-parameter configuration suggested by the authors of each VT. However, the hyper-parameters are chosen based on the specific datasets and tasks in the original paper, which might not be appropriate for all the datasets used in this paper. The fact that the top-1 accuracy of models trained on certain datasets is extremely low (e.g. Infograph) also suggests that. Knowing that it is quite computationally expensive and unnecessary to tune every single model, I would suggest to lightly tune once for each dataset to make sure the performance does not largely influenced by wrongly chosen hyper-parameters.

Thirdly, the reason why the paper uses DomainNet datasets is not very clear to me. Hence, it is also not so clear that the comparisons of experimental results between training from scratch and fine-tuning is necessary. The paper gives its reason for using DomainNet datasets: "We chose the latter because of the large domain-shift between some of its datasets and ImageNet, which makes the fine-tuning experiments non-trivial," but it is not articulated how would this results support the two major claims of the paper in the introduction. If the paper is trying to show that certain architecture is more robust in training small-medium sized datasets, then it is quite indirect to use domainNet datasets for this purpose.

Overall, the method proposed by this paper seems effective and can be used in the future by other researchers. However, it still needs more careful experimental design to support and solidify this method's effectiveness.



**Time Spent Reviewing:**

6

---

> ### Author Response · Authors · 2021-08-10
> **Answers to Reviewer 44YH**
>
> **1.Q. The novelty of this design is minor. … the method is not very different from ELECTRA. A.** We disagree that the originality of our proposal is minor, since, as far as we know, we are the first proposing a mixed supervised/unsupervised learning paradigm to regularize VT training with scarcity of training data/computational resources. Moreover, our task is different from the one proposed in ELECTRA [11], and can be seen as a geometric version of the ELECTRA task. Indeed, in ELECTRA, the original sentence tokens are randomly replaced by tokens generated by another (pre-trained) transformer (e.g., a small “BERT”-like “generator”). The task is to guess what token is original and what was replaced. Conversely, in our case, we do not replace input tokens. The reason is that in NLP tasks, tokens are discrete and limited (e.g., the set of words of a specific-language dictionary), while image patches are “continuous” and highly variable, so a replacement-based task is hard to use in computer vision. In contrast, we propose to predict the spatial distance of densely, randomly selected token embedding pairs. Thus, our task is based on “geometry”, while the ELECTRA task is based on a discrete dictionary. Note also that we do not need a second pretrained transformer (the “generator” used in [11] to propose plausible replacements).
>
> **2.Q. It is a bit unnecessary for the paper to spend much space talking about those loss variants that do not actually perform well. A.** We believe that the interest of these variants lies in discussing different implementations of our pretext task which can guide people interested in reproducing our method (e.g., avoiding less useful directions). Moreover, the analysis of their relative performance (l. 287-293 and Supplementary Material, l. 58-63) contributes to understand what are the important ingredients of our task. Finally, please note that all the proposed variants except the last (the “very dense localization loss”) outperform the baseline in Tab. 1 (b) (especially the cross-entropy based loss), thus they are not that low-performing.
>
> **3.Q. Is the improvement statistically significant? A.** Unfortunately, using different runs (with different random seeds) for all the 22 experiments of Tab. 2 (b), Tab. 3 and Tab. 4 (plus those included in the Supplementary Material), with 6 models (3 baselines plus our 3 corresponding VTs) is too computationally demanding. However, we believe that, since we have observed an improvement in **all** these experiments and with **all** the tested VTs (sometimes a dramatic improvement), our conclusions are statistically robust. Nevertheless, following your suggestion, we repeated the experiments on ImageNet-100 (Tab. 2 (b)) using 5 runs with 5 different random seeds (and training for 100 epochs). The table below shows the results, averaged over the 5 runs (+- the standard deviation):
>
> |Model|Original|+ L_{drloc}|
> |:----|:----:|:----:|
> |CvT-13|85.62+-0.05|86.09+-0.12|
> |Swin-T|82.66+-0.10|83.95+-0.05|
> |T2T-ViT-14|82.67+-0.01|83.74+-0.08|
>
> Table: Statistical significance on ImageNet-100.
>
> These results confirm those reported in Tab. 2 (b), and in the final version we will replace Tab. 2 (b) with the values here reported.
>
> **4.Q. hyper-parameter configuration tuned for each dataset. A.** We agree that tuning the hyper-parameters for each specific dataset and task (training from scratch, finetuning, object detection, segmentation, etc.) can boost the performance of each method. However, this is unfortunately undoable because too computationally demanding (please, see the answer above). We selected the Infograph dataset, and we tuned what is probably the most important hyper-parameter, i.e., the learning rate. We report below the results (each column corresponds to a different learning rate):
>
> |Method|0.005|0.001(default)|0.0005|
> |:----|:----:|:----:|:----:|
> |CvT-13|3.25|19.39|19.85|
> |CvT-13 + L_{drloc}|3.51|20.05|20.40|
> |Swin-T|2.93|8.20|13.02|
> |Swin-T + L_{drloc}|3.01|10.16|13.62|
> |T2T-ViT|2.57|6.89|13.59|
> |T2T-ViT + L_{drloc}|2.57|9.51|14.93|
>
> Table: Ablation study on the base learning rate on Infograph.
>
> Apart from T2T and learning rate 0.005 (where our results are on par with the baseline), **in all the other configurations, we always improve the corresponding baseline**, which we believe further confirms the robustness of our approach.
>
> **5.Q. Why the paper uses DomainNet datasets. A.** In Tab. 4, if you compare the results of T2T when finetuned on CIFAR-10 (98.37) with the results obtained when T2T is finetuned on Infograph (38.53), you see that there is a difference of about 60 points. This shows that the largest domain-shift (w.r.t. ImageNet-1k) characterizing Infograph, can make a big difference in the VT finetuning accuracy. On the other hand, the advantage of our regularization task is more prominent in the latter case (+7.16 on Infograph) than in the former dataset (+0.15). This example explains why we chose to use also the datasets of DomainNet, most of which are characterized by a large domain-shift w.r.t. ImageNet: we wanted to test both the VT baselines and our method in a scenario in which pre-training on ImageNet is not sufficient to achieve a good performance. We will add a deeper explanation on this point in the revised version.
>
> **6.Q. Quantifying the differences in accuracies among different VT architectures. A.** This analysis is actually contained in l. 310-317. We will add additional details on this point on the Supplementary Material (please, consider that in the main paper we will need to add other experimental results suggested by all the Reviewers).

---

### Decision · Program_Chairs · 2021-09-28

**Decision:**

Accept (Poster)

**Comment:**

I enjoyed reading this paper and I find it generally solid. I find that the most important element of this paper is a new, presumably novel, self-supervised learning task that regularizes the vision transformer. Although the results presented in the paper were reasonably strong, the reviewers found several issues in the experiments. The authors provided answers which unfortunately did not satisfy the reviewers enough to increase their scores. Although this paper is not very far from meeting the bar for acceptance, I think it is still slightly below the expected level. Some things that in my opinion could tip the balance in the next version: (1) improved experiments, (2) a better justification for the proposed self-supervised task.

**Consistency Experiment:**

NeurIPS has a long history of experimentation. In 2014, NeurIPS ran an experiment in which 10% of submissions were reviewed by two independent committees to quantify the randomness in the review process. This year, we repeated a variant of this experiment to see how the quality of the review process has changed over time.  This paper was part of the experiment and was therefore assigned to two committees (consisting of reviewers, an Area Chair, and a Senior Area Chair) that reached independent decisions.  If both committees made the same recommendation, this recommendation was followed. If a single committee recommended acceptance, the paper was accepted (with the exception of a few cases in which the other committee identified what we considered a fatal flaw, e.g., an error in a key result).

This copy’s committee reached the following decision: **Reject**

The other committee assigned to the paper recommended **Accept (Spotlight)**.  You can find the other set of reviews, along with any follow up discussion with the authors here:
https://openreview.net/forum?id=AJofO-OFT40